# Meta-Learning to Improve Pre-Training

**Aniruddh Raghu**
Massachusetts Institute of Technology
`araghu@mit.edu`

**Jonathan Lorraine**
University of Toronto

**Simon Kornblith**
Google Research

**Matthew McDermott**
Massachusetts Institute of Technology

**David Duvenaud**
Google Research & University of Toronto

## Abstract

Pre-training (PT) followed by fine-tuning (FT) is an effective method for training neural networks, and has led to significant performance improvements in many domains. PT can incorporate various design choices such as task and data reweighting strategies, augmentation policies, and noise models, all of which can significantly impact the quality of representations learned. The hyperparameters introduced by these strategies therefore must be tuned appropriately. However, setting the values of these hyperparameters is challenging. Most existing methods either struggle to scale to high dimensions, are too slow and memory-intensive, or cannot be directly applied to the two-stage PT and FT learning process. In this work, we propose an efficient, gradient-based algorithm to meta-learn PT hyperparameters. We formalize the PT hyperparameter optimization problem and propose a novel method to obtain PT hyperparameter gradients by combining implicit differentiation and backpropagation through unrolled optimization. We demonstrate that our method improves predictive performance on two real-world domains. First, we optimize high-dimensional task weighting hyperparameters for multitask pre-training on protein-protein interaction graphs and improve AUROC by up to 3.9%. Second, we optimize a data augmentation neural network for self-supervised PT with SimCLR on electrocardiography data and improve AUROC by up to 1.9%.

## 1 Introduction

A popular and important learning paradigm for neural networks is pre-training (PT) followed by fine-tuning (FT), an approach commonly used in transfer learning [13, 59, 19, 27, 52, 11, 37, 74, 35, 28], and semi-supervised learning [9, 8, 24]. This paradigm has led to performance improvements in many domains, including computer vision [13, 59, 19, 37, 74, 35], natural language processing [27, 52, 11, 40, 34], graph structured prediction [28], and clinical machine learning [45, 46, 2, 48], and is especially helpful in settings where downstream tasks have limited training data.

The PT & FT paradigm introduces high-dimensional, complex PT hyperparameters, such as parameterized data augmentation policies used in contrastive representation learning [8, 22] or the use of task, class, or instance weighting variables in multi-task PT to avoid negative transfer [70]. These hyperparameters can significantly affect the quality of pre-trained models [8], and thus finding techniques to set their values optimally is an important area of research.

Choosing optimal PT hyperparameter values is challenging, and existing methods do not work well. Simple approaches such as random or grid search are inefficient since evaluating a hyperparameter setting requires performing the full, two-stage PT & FT optimization, which may be prohibitively computationally expensive. Gradient-free approaches, such as Bayesian optimization or evolutionary algorithms [33, 61, 47], are also limited in how well they scale to this setting. Gradient-based

35th Conference on Neural Information Processing Systems (NeurIPS 2021).

approaches [44, 41, 43, 42] can be used online to jointly learn hyperparameters and model parameters and can scale to millions of hyperparameters [42], but typically deal with a standard *single-stage* learning problem (e.g., normal supervised learning) and are therefore not directly applicable to the *two-stage* PT & FT learning problem.

In this work, we address this gap and propose a method for high-dimensional PT hyperparameter optimization. We first formalize a variant of the PT & FT paradigm, which we call *meta-parameterized pre-training* (Figure 1), where *meta-parameters* refer to arbitrary PT hyperparameters or parameterizable architectural choices that can be optimized to improve the learned representations.[1] We outline a meta-learning problem characterizing the optimal meta-parameters propose a gradient-based method to learn meta-parameters. Our contributions are:

- We formalize *meta-parameterized pre-training*, a variant of the pre-training and fine-tuning (PT & FT) paradigm where PT is augmented to incorporate *meta-parameters*: arbitrary structures that can be optimized to improve learned representations.
- We propose a scalable gradient-based algorithm to learn meta-parameters using a novel method to obtain meta-parameter gradients through the two-stage PT & FT process. Our gradient estimator composes a constant-memory implicit differentiation approximation for the longer PT stage and exact backpropagation through training for the shorter FT stage.
- We show that our algorithm recovers optimal meta-parameters in toy experiments on synthetic data.
- In two real-world experimental domains, we demonstrate our algorithm improves performance. Firstly, on a multitask PT benchmark over biological graph-structured data [28], using our method to optimize meta-parameters representing task weights improves performance by up to 3.9% AUROC. Secondly, for semi-supervised learning using SimCLR [8] over electrocardiography data, using our algorithm to optimize meta-parameters representing the weights of a data augmentation neural network improves performance by up to 1.9% AUROC.

## 2 Problem Setup and Preliminaries

In this section, we define the meta-parameterized pre-training meta-learning problem, and compare it to traditional fine-tuning and pre-training. A full glossary of notation is in Appendix B, Table 3.

**Notation.** Let the subscript $\bullet$ be a placeholder for either PT (pre-training) or FT (fine-tuning), $\mathcal{X} \subseteq \mathbb{R}^d$ be our input domain, $\mathcal{Y}_\bullet$ and $\hat{\mathcal{Y}}_\bullet$ be the true and predicted output spaces for some model respectively, and $\Theta, \Psi_\bullet, \Phi$ be spaces of parameters for models. We will use $f_\bullet : \mathcal{X}; (\Theta, \Psi_\bullet) \to \hat{\mathcal{Y}}_\bullet$ to refer to a parametric model, with the semicolon separating the input space from the parameter spaces. We then define $f_\bullet = f_\bullet^{(\text{head})} \circ f^{(\text{feat})}$, such that $f^{(\text{feat})}(\cdot; \boldsymbol{\theta} \in \Theta)$ is a *feature extractor* that is transferable across learning stages (e.g., pre-training to fine-tuning), and $f_\bullet^{(\text{head})}(\cdot; \boldsymbol{\psi} \in \Psi_\bullet)$ is a stage-specific *head* that is not transferable. Given a data distribution $\mathbf{x}_\bullet, \mathbf{y}_\bullet \sim \mathcal{D}_\bullet$, parametric model $f_\bullet$, and loss function $\mathcal{L}_\bullet : \hat{\mathcal{Y}}_\bullet \times \mathcal{Y}_\bullet \to \mathbb{R}$, we will also define for convenience a corresponding expected loss $L_\bullet : \Theta, \Psi_\bullet \to \mathbb{R}$ via $L_\bullet(\boldsymbol{\theta}, \boldsymbol{\psi}_\bullet; \mathcal{D}_\bullet) = \mathbb{E}_{\mathcal{D}_\bullet}[\mathcal{L}_\bullet(f_\bullet(\boldsymbol{x}_\bullet; \boldsymbol{\theta}, \boldsymbol{\psi}_\bullet), y_\bullet)]$. We also adopt the convention that the output of the argmin operator is *any* arbitrary minimum, rather than the set of possible minima, to avoid complications in notation.

### 2.1 Problem Formulation

**Supervised Learning (Fig. 1A).** In a fully-supervised setting (our *fine-tuning* domain), we are given a data distribution $\mathcal{D}_{\text{FT}}$, model $f$, and loss $\mathcal{L}_{\text{FT}}$. Using a *learning algorithm* $\text{Alg}_{\text{FT}}$ (e.g., SGD) that takes as input initial parameters $\boldsymbol{\theta}_{\text{FT}}^{(0)}, \boldsymbol{\psi}_{\text{FT}}^{(0)}$, our goal is to approximate the $\mathcal{L}_{\text{FT}}$-optimal parameters:
$\boldsymbol{\theta}_{\text{FT}}^*, \boldsymbol{\psi}_{\text{FT}}^* = \text{Alg}_{\text{FT}}(\boldsymbol{\theta}_{\text{FT}}^{(0)}, \boldsymbol{\psi}_{\text{FT}}^{(0)}; \mathcal{D}_{\text{FT}}) \approx \text{argmin}_{\boldsymbol{\theta} \in \Theta, \boldsymbol{\psi} \in \Psi_{\text{FT}}} L_{\text{FT}}(\boldsymbol{\theta}, \boldsymbol{\psi}; \mathcal{D}_{\text{FT}})$

**Pre-training (Fig. 1B).** For tasks where data is scarce, we can additionally incorporate a *pre-training* step and approximate the optimal initial parameters for FT (i.e., the final pre-trained weights are used as initialization weights of the FT stage), again via an optimization algorithm $\text{Alg}_{\text{PT}}$:
$\boldsymbol{\theta}_{\text{PT}}^* = \text{Alg}_{\text{PT}}(\boldsymbol{\theta}_{\text{PT}}^{(0)}, \boldsymbol{\psi}_{\text{PT}}^{(0)}; \mathcal{D}_{\text{PT}}) \approx \text{argmin}_{\boldsymbol{\theta} \in \Theta} L_{\text{FT}}(\text{Alg}_{\text{FT}}(\boldsymbol{\theta}, \boldsymbol{\psi}_{\text{FT}}^{(0)}; \mathcal{D}_{\text{FT}}); \mathcal{D}_{\text{FT}})$. [2]

---

[1] We use the term *meta-parameter* since these structures do not directly affect inference of the final model after FT, but instead inform the process of learning this model (by modulating the PT process).

[2] Note that we discard the PT head $\boldsymbol{\psi}_{\text{PT}}^*$ here as only the PT feature extractor $\boldsymbol{\theta}_{\text{PT}}^*$ is transferred.

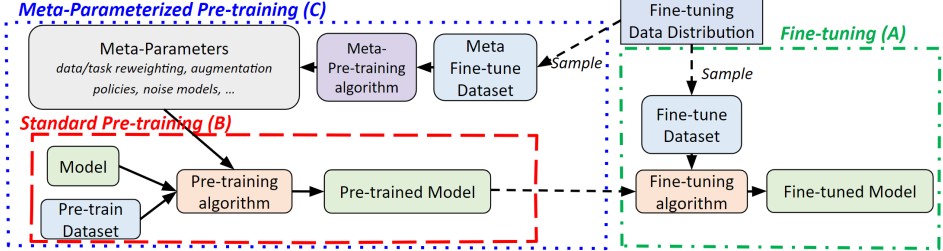

Figure (1)   **Meta-Parameterized Pre-Training.** A paradigm where meta-parameters — rich, potentially high dimensional structures that generalize PT hyperparameters — are incorporated in PT to improve the learned representations. Meta-parameters are optimized in a *meta-PT* phase, using data from FT task(s) in a meta-FT dataset. The FT and meta-FT datasets are (potentially overlapping) samples from the FT data distribution.

**Meta-Parameterized PT (Fig. 1C).** In *Meta-Parameterized PT*, we recognize that, in addition to taking as input the PT parameters $\boldsymbol{\theta}$, $\text{Alg}_{\text{PT}}$ is itself parameterized by a set of *meta-parameters* $\boldsymbol{\phi} \in \Phi$: arbitrary, potentially high dimensional quantities that inform the structure of the algorithm directly. These could represent weighting strategies, data augmentation policies, or sampling processes. The optimal meta-parameters $\boldsymbol{\phi}^{(\text{opt})}$ are the solution to the following *meta-PT* optimization problem:

$$\boldsymbol{\phi}^{(\text{opt})} = \underset{\boldsymbol{\phi} \in \Phi}{\text{argmin}}\, L_{\text{FT}} \left( \text{Alg}_{\text{FT}} \left( \text{Alg}_{\text{PT}} \left( \boldsymbol{\theta}_{\text{PT}}^{(0)}, \boldsymbol{\psi}_{\text{PT}}^{(0)}; \mathcal{D}_{\text{PT}}, \boldsymbol{\phi} \right), \boldsymbol{\psi}_{\text{FT}}^{(0)}; \mathcal{D}_{\text{FT}} \right); \mathcal{D}_{\text{FT}} \right).$$

## 2.2   Example: Multitask Meta-Parameterized Pre-Training

To make our notation concrete, here we instantiate our setup for a multitask pre-training problem.

**Problem:**   Suppose we have a multitask classification dataset, $(\mathcal{X} \times \mathcal{Y})^N$ such that $\mathcal{Y} = \mathcal{Y}_1 \times \cdots \times \mathcal{Y}_K$ consists of labels for $K$ distinct tasks. Of this full set of tasks, we are interested only in a subset of $M$ tasks, $S = \{t_1, \ldots, t_M\} \subseteq \{1, \ldots, K\}$.
**Supervised FT:** Under supervised FT alone, we can directly average a cross-entropy loss $\mathcal{L}_{\text{CE}}$ over *only the tasks in* $S$, $\mathcal{L}_{\text{FT}}(\hat{\boldsymbol{y}}, \boldsymbol{y}) = \frac{1}{M} \sum_{j=1}^{M} \mathcal{L}_{\text{CE}}(\hat{y}^{(t_j)}, y^{(t_j)})$, and then solve this problem via SGD.
**PT:** If we assume that $S$ is a *random* subset of the full set of tasks, we can introduce a PT stage over all tasks: $\mathcal{L}_{\text{PT}}(\hat{\boldsymbol{y}}, \boldsymbol{y}) = \frac{1}{K} \sum_{i=1}^{K} \mathcal{L}_{\text{CE}}(\hat{y}^{(i)}, y^{(i)})$, followed by FT on $S$ alone. As $S$ is a random subset, leveraging all tasks for PT is well motivated and may improve performance.
**Meta-Parameterized PT:** In the case where $T$ is not a random subset, the PT strategy described above is no longer well-motivated. However, using meta-parameterized PT, we can still effectively pre-train by introducing the meta-parameters that weight the tasks $\boldsymbol{\phi} = [\phi_1 \quad \ldots \quad \phi_K]$ and modulate the loss function $\mathcal{L}_{\text{PT}}$: $\mathcal{L}_{\text{PT}}(\hat{\boldsymbol{y}}, \boldsymbol{y}; \boldsymbol{\phi}) = \sum_{i=1}^{K} \phi_i \mathcal{L}_{CE}(\hat{y}^{(i)}, y^i)$. With optimal meta-parameters $\boldsymbol{\phi}^{(\text{opt})}$, the PT stage will leverage only that subset of tasks that best informs the final FT performance. This setting mirrors our real-world experiment in Section 5.

## 3   Methods: Optimizing Meta-Parameters for Two-Stage Training

We now introduce our gradient-based algorithm to optimize meta-parameters. We first describe how to efficiently approximate meta-parameter gradients through the two-stage PT and FT optimization. We then present our algorithm, and outline practical considerations when using it.

### 3.1   Efficient Computation of Meta-Parameter Gradients

We begin by defining:

$$g(\boldsymbol{\phi}; \boldsymbol{\theta}_{\text{PT}}^{(0)}, \boldsymbol{\psi}_{\text{PT}}^{(0)}, \boldsymbol{\psi}_{\text{FT}}^{(0)}) = L_{\text{FT}} \Big( \underbrace{\text{Alg}_{\text{FT}} \big( \overbrace{\text{Alg}_{\text{PT}}(\boldsymbol{\theta}_{\text{PT}}^{(0)}, \boldsymbol{\psi}_{\text{PT}}^{(0)}; \mathcal{D}_{\text{PT}}, \boldsymbol{\phi})}^{\text{Parameter } \boldsymbol{\theta}_{\text{PT}}}, \boldsymbol{\psi}_{\text{FT}}^{(0)}; \mathcal{D}_{\text{FT}} \big)}_{\text{Parameters } \boldsymbol{\theta}_{\text{FT}}, \boldsymbol{\psi}_{\text{FT}}}; \mathcal{D}_{\text{FT}} \Big), \qquad (1)$$

so that $\boldsymbol{\phi}^{(\text{opt})} = \text{argmin}_{\boldsymbol{\phi} \in \Phi}\, g(\boldsymbol{\phi})$.

We also define two best-response values:

$$\boldsymbol{\theta}_{\mathrm{PT}}^*(\boldsymbol{\phi}) = \mathrm{Alg}_{\mathrm{PT}}(\boldsymbol{\theta}_{\mathrm{PT}}^{(0)}, \boldsymbol{\psi}_{\mathrm{PT}}^{(0)}; \mathcal{D}_{\mathrm{PT}}, \boldsymbol{\phi}),$$

$$\boldsymbol{\theta}_{\mathrm{FT}}^*(\boldsymbol{\phi}), \boldsymbol{\psi}_{\mathrm{FT}}^*(\boldsymbol{\phi}) = \mathrm{Alg}_{\mathrm{FT}}(\boldsymbol{\theta}_{\mathrm{PT}}^*(\boldsymbol{\phi}), \boldsymbol{\psi}_{\mathrm{FT}}^{(0)}; \mathcal{D}_{\mathrm{FT}}).$$

We do not explicitly include the dependence of the best responses on the initialization values for notational convenience.

With these defined, we now consider the desired gradient term, $\frac{\partial g}{\partial \phi}$. Under our definitions, the direct partial derivatives $\frac{\partial L_{\mathrm{FT}}}{\partial \phi}$ and $\frac{\partial \mathrm{Alg}_{\mathrm{FT}}}{\partial \phi}$ are zero, so $\frac{\partial g}{\partial \phi}$ reduces to a simple expression of the chain rule:

$$\left.\frac{\partial g}{\partial \phi}\right|_{\phi'} = \underbrace{\left.\frac{\partial L_{\mathrm{FT}}}{\partial [\boldsymbol{\theta}_{\mathrm{FT}}, \quad \boldsymbol{\psi}_{\mathrm{FT}}]}\right|_{\boldsymbol{\theta}_{\mathrm{FT}}^*(\phi'), \boldsymbol{\psi}_{\mathrm{FT}}^*(\phi')}}_{\text{FT Loss Gradient}} \times \overbrace{\left.\frac{\partial \mathrm{Alg}_{\mathrm{FT}}}{\partial \boldsymbol{\theta}_{\mathrm{PT}}}\right|_{\boldsymbol{\theta}_{\mathrm{PT}}^*(\phi')}}^{\text{FT Best Response Jacobian}} \times \underbrace{\left.\frac{\partial \mathrm{Alg}_{\mathrm{PT}}}{\partial \phi}\right|_{\phi'}}_{\text{PT Best Response Jacobian}}. \tag{2}$$

The FT Loss Gradient term on the RHS of (2) is easily computed using backpropagation. Computing the other two terms is more involved, and we detail each below, beginning with the PT best response Jacobian. The full algorithm with both gradient estimation terms is provided in Algorithm 1.

**PT Best Response Jacobian** $\frac{\partial \mathrm{Alg}_{\mathrm{PT}}}{\partial \phi}$. Using recent work in hyperparameter optimization with implicit differentiation [42], we re-express this term using the implicit function theorem (IFT). If we assume that $\boldsymbol{\theta}_{\mathrm{PT}}^*(\boldsymbol{\phi}) = \mathrm{Alg}_{\mathrm{PT}}\left(\boldsymbol{\theta}_{\mathrm{PT}}^{(0)}; \mathcal{D}_{\mathrm{PT}}, \boldsymbol{\phi}\right)$ is a good approximation of $\mathrm{argmin}_{\boldsymbol{\theta} \in \Theta} L_{\mathrm{PT}}(\boldsymbol{\theta}; \mathcal{D}_{\mathrm{PT}}, \boldsymbol{\phi})$ (i.e., the PT model converges to $\mathcal{L}_{\mathrm{PT}}$-optimal parameters), then under certain smoothness and regularity assumptions on the PT parameters and meta-parameters, the IFT allows us to re-express $\frac{\partial \mathrm{Alg}_{\mathrm{PT}}}{\partial \phi}$ as:

$$\left.\frac{\partial \mathrm{Alg}_{\mathrm{PT}}}{\partial \phi}\right|_{\phi'} = -\left[\frac{\partial^2 L_{\mathrm{PT}}}{\partial \boldsymbol{\theta}_{\mathrm{PT}} \partial \boldsymbol{\theta}_{\mathrm{PT}}^\top}\right]^{-1} \times \left.\frac{\partial^2 L_{\mathrm{PT}}}{\partial \boldsymbol{\theta}_{\mathrm{PT}} \partial \phi^\top}\right|_{\boldsymbol{\theta}_{\mathrm{PT}}^*(\phi'), \phi'}, \tag{3}$$

which is the product of the inverse Hessian and a matrix of mixed partial derivatives. Following [42], the inverse can be efficiently approximated using a truncated Neumann series.

**FT Best Response Jacobian** $\frac{\partial \mathrm{Alg}_{\mathrm{FT}}}{\partial \boldsymbol{\theta}_{\mathrm{PT}}}$. First, note that without additional constraints on $\mathrm{Alg}_{\mathrm{FT}}$, the FT best response Jacobian may be zero. This is because $L_{\mathrm{FT}}$ has no functional dependence on the variable $\boldsymbol{\theta}_{\mathrm{PT}}$ and, if we assume the convergence point $\boldsymbol{\theta}_{\mathrm{FT}}^*$ is stable (as we did for the PT best response Jacobian), this implies that the gradient of $\boldsymbol{\theta}_{\mathrm{FT}}^*$ with respect to $\boldsymbol{\theta}_{\mathrm{PT}}$ would be zero. To enable effective learning, we must therefore either (1) impose restrictions on $\mathrm{Alg}_{\mathrm{FT}}$ to ensure there is a dependence between the initialization point and the final loss value (e.g., proximal regularization [55]) or (2) leverage methods that do not differentiate through $\mathrm{Alg}_{\mathrm{FT}}$ through convergence, as at non-converged points we will still observe nonzero $L_{\mathrm{FT}}$-gradients [29, 51]. Given that the FT phase often involves shorter optimization horizons than PT, we take approach 2 here, and iteratively update $\boldsymbol{\theta}_{\mathrm{FT}}$ for $K$ steps. We first initialize the FT head $\boldsymbol{\psi}_{\mathrm{FT}}^{(0)}$ and then compute:

$$\boldsymbol{\theta}_{\mathrm{FT}}^{(0)} = \mathrm{copy}(\boldsymbol{\theta}_{\mathrm{PT}}^*) \qquad \text{(init with PT solution, implicitly performing stop gradient)}$$

$$\boldsymbol{\theta}_{\mathrm{FT}}^{(k)}, \boldsymbol{\psi}_{\mathrm{FT}}^{(k)} = \left[\boldsymbol{\theta}_{\mathrm{FT}}^{(k-1)}, \quad \boldsymbol{\psi}_{\mathrm{FT}}^{(k-1)}\right] - \eta_{\mathrm{FT}} \left.\frac{\partial L_{\mathrm{FT}}}{\partial [\boldsymbol{\theta}_{\mathrm{FT}}, \quad \boldsymbol{\psi}_{\mathrm{FT}}]}\right|_{\boldsymbol{\theta}_{\mathrm{FT}}^{(k-1)}, \boldsymbol{\psi}_{\mathrm{FT}}^{(k-1)}} \qquad k = 1, \ldots, K \tag{4}$$

$$\boldsymbol{\theta}_{\mathrm{FT}}^*, \boldsymbol{\psi}_{\mathrm{FT}}^* \approx \boldsymbol{\theta}_{\mathrm{FT}}^{(K)}, \boldsymbol{\psi}_{\mathrm{FT}}^{(K)},$$

and compute the gradient $\left.\frac{\partial \mathrm{Alg}_{\mathrm{FT}}}{\partial \boldsymbol{\theta}_{\mathrm{PT}}}\right|_{\boldsymbol{\theta}_{\mathrm{PT}}^*(\phi')}$ by differentiating through this optimization.[3]

We can also choose to freeze the feature extractor parameters $\boldsymbol{\theta}_{\mathrm{FT}}$ and update only the head parameters $\boldsymbol{\psi}_{\mathrm{FT}}$ during truncated FT, and use this to obtain meta-parameter gradients. This resembles *linear evaluation*, where a linear classifier is trained on top of fixed, pre-trained feature extractors [50, 3, 63].

Together, these two approximations allow for efficient computation of meta-parameter gradients.

---

[3]While Equation 4 uses standard gradient descent, we could use other differentiable optimizers (e.g., Adam).

**Algorithm 1** Gradient-based algorithm to learn meta-parameters. Notation defined in Appendix B, Table 3. Vector-Jacobian products (VJPs) can be efficiently computed by standard autodifferentiation.

1: Initialize PT parameters $\boldsymbol{\theta}_{\text{PT}}^{(\text{init})}, \boldsymbol{\psi}_{\text{PT}}^{(\text{init})}, \boldsymbol{\psi}_{\text{FT}}^{(0)}$ and meta-parameters $\boldsymbol{\phi}^{(0)}$
2: **for** $n = 1, \ldots, N$ iterations **do**
3:     Initialize $\boldsymbol{\theta}_{\text{PT}}^{(0)} = \boldsymbol{\theta}_{\text{PT}}^{(\text{init})}$ and $\boldsymbol{\psi}_{\text{PT}}^{(0)} = \boldsymbol{\psi}_{\text{PT}}^{(\text{init})}$.
4:     **for** $p = 1, \ldots, P$ PT iterations **do**
5:         $\left[\boldsymbol{\theta}_{\text{PT}}^{(p)}, \boldsymbol{\psi}_{\text{PT}}^{(p)}\right] = \left[\boldsymbol{\theta}_{\text{PT}}^{(p-1)}, \boldsymbol{\psi}_{\text{PT}}^{(p-1)}\right] - \eta_{\text{PT}} \left. \frac{\partial L_{\text{PT}}}{\partial[\boldsymbol{\theta}_{\text{PT}}, \boldsymbol{\psi}_{\text{PT}}]} \right|_{\boldsymbol{\theta}_{\text{PT}}^{(p-1)}, \boldsymbol{\psi}_{\text{PT}}^{(p-1)}}$
6:     **end for**
7:     Initialize FT encoder with PT solution: $\boldsymbol{\theta}_{\text{FT}}^{(0)} = \texttt{copy}(\boldsymbol{\theta}_{\text{PT}}^{(P)})$.
8:     Approximate $\boldsymbol{\theta}_{\text{FT}}^*, \boldsymbol{\psi}_{\text{FT}}^*$ using Eq. 4.
9:     Compute $g_1 = \left. \frac{\partial L_{\text{FT}}}{\partial[\boldsymbol{\theta}_{\text{FT}}, \boldsymbol{\psi}_{\text{FT}}]} \right|_{\boldsymbol{\theta}_{\text{FT}}^*, \boldsymbol{\psi}_{\text{FT}}^*}$
10:    Compute VJP $g_2 = g_1 \left. \frac{\partial \text{Alg}_{\text{FT}}}{\partial \boldsymbol{\theta}_{\text{PT}}} \right|_{\boldsymbol{\theta}_{\text{PT}}^{(P)}, \boldsymbol{\psi}_{\text{FT}}^{(0)}}$ using the unrolled learning step from line 8.
11:    Approximate VJP $\left. \frac{\partial g}{\partial \boldsymbol{\phi}} \right|_{\boldsymbol{\phi}^{(n-1)}} = g_2 \left. \frac{\partial \text{Alg}_{\text{PT}}}{\partial \boldsymbol{\phi}} \right|_{\boldsymbol{\phi}^{(n-1)}}$ using the IFT (Eq. 3).
12:    $\boldsymbol{\phi}^{(n)} = \boldsymbol{\phi}^{(n-1)} - \eta_{\text{V}} \left. \frac{\partial g}{\partial \boldsymbol{\phi}} \right|_{\boldsymbol{\phi}^{(n-1)}}$
13:    Update PT initialization by setting: $\boldsymbol{\theta}_{\text{PT}}^{(\text{init})} = \boldsymbol{\theta}_{\text{PT}}^{(P)}$ and $\boldsymbol{\psi}_{\text{PT}}^{(\text{init})} = \boldsymbol{\psi}_{\text{PT}}^{(P)}$.
14: **end for**

## 3.2 Our Algorithm and Practical Considerations

By leveraging the above approximations, we obtain Algorithm 1 to optimize meta-parameters $\boldsymbol{\phi}$ online during PT & FT of the base model. Note that $\text{Alg}_{\text{PT}}$ is explicitly written out as a sequence of gradient updates (lines 4-6 in Algorithm 1). We now discuss practical considerations when using this algorithm, with further details given in Appendix C.

**(1) Access to $\mathcal{D}_{\text{FT}}$ and generalizing to new FT tasks:** Solving the meta-PT problem requires availability of: the model $f_\bullet$, the PT data $\mathcal{D}_{\text{PT}}$, and the FT data $\mathcal{D}_{\text{FT}}$. In this work, we assume availability of the model and PT dataset, but since assuming access to the complete FT dataset at meta-PT time is more restrictive, we study two scenarios: *Full FT Access*, where all FT data that we expect to encounter is available at meta-PT time, and *Partial FT Access*, where the FT data available at meta-PT time is only a sample from a distribution of FT data that we may encounter later.

*Full FT Access* occurs in settings like semi-supervised learning, where we are given a large unlabelled PT dataset and a small labelled FT dataset and our goal is to achieve the best possible performance by leveraging these two fixed datasets [68, 73, 25, 24, 8, 9].

*Partial FT Access* occurs when our goal is to learn transferable representations: at meta-PT time, we might have limited knowledge of FT tasks or data. In evaluating this scenario, we examine generalizability to new FT tasks, given only small amounts of FT data/task availability at meta-PT time, demonstrating that even very limited FT access can be sufficient for effective meta-parameter optimization [11, 45, 56, 28].

**(2) $\mathcal{D}_{\text{FT}}$ splits:** In practice, we have access to finite datasets and use minibatches, rather than true data-generating processes. Following standard convention, we split $\mathcal{D}_{\text{FT}}$ into two subsets for meta-learning: $\mathcal{D}_{\text{FT}}^{(\text{tr})}$ and $\mathcal{D}_{\text{FT}}^{(\text{val})}$ (independent of any held-out $\mathcal{D}_{\text{FT}}$ testing split), and define the FT data available at meta-PT time as $\mathcal{D}_{\text{FT}}^{(\text{Meta})} = \mathcal{D}_{\text{FT}}^{(\text{tr})} \cup \mathcal{D}_{\text{FT}}^{(\text{val})}$. We use $\mathcal{D}_{\text{FT}}^{(\text{tr})}$ for the computation of $\left. \frac{\partial \text{Alg}_{\text{FT}}}{\partial \boldsymbol{\theta}_{\text{PT}}} \right|_{\boldsymbol{\theta}_{\text{PT}}^{(P)}, \boldsymbol{\psi}_{\text{FT}}^{(0)}}$ and $\left. \frac{\partial \text{Alg}_{\text{PT}}}{\partial \boldsymbol{\phi}} \right|_{\boldsymbol{\phi}^{(n-1)}}$ and $\mathcal{D}_{\text{FT}}^{(\text{val})}$ for the computation of $\left. \frac{\partial L_{\text{FT}}}{\partial[\boldsymbol{\theta}_{\text{FT}}, \boldsymbol{\psi}_{\text{FT}}]} \right|_{\boldsymbol{\theta}_{\text{FT}}^*, \boldsymbol{\psi}_{\text{FT}}^*}$ in Algorithm 1.

**(3) Online updates:** Given that PT phases often involve long optimization horizons, for computational efficiency, we update $\boldsymbol{\theta}_{\text{PT}}$ and $\boldsymbol{\psi}_{\text{PT}}$ online rather than re-initializing them at every meta-iteration (see Algorithm 1). FT phases are often shorter so we could in theory re-initialize $\boldsymbol{\psi}_{\text{FT}}$ at each

meta-iteration, as is presented in Algorithm 1. However, it is more computationally efficient to also optimize this online, and we follow this approach in our experiments. A description of the algorithm with these details in Appendix C.

Note that prior work [67] has suggested that online optimization of certain hyperparameters (e.g., learning rates) using short horizons may yield suboptimal solutions. We comment on this in Appendix C, study this effect for our algorithm in synthetic experiments in Appendix E, and in real-world experiments on self-supervised learning in Appendix G, revealing it is not a significant concern.

**(4) Computational tractability:** Our method can scale to large encoder models and high-dimensional meta-parameters, despite the complexity of the two-stage PT & FT process. This is because: (i) meta-parameters are optimized jointly with the base model parameters; (ii) using the IFT to obtain gradients has similar time and memory complexity to one iteration of training [42]; (iii) the FT best response Jacobian can be approximated efficiently using a small number of unrolled optimization steps $K$, and by only unrolling the FT head of the network. In our real-world experiments (Sections 5 and 6), meta-parameterized PT has less than twice the time cost of standard PT. Further details on time and memory cost are provided in Appendices F and G.

**(5) Setting optimizer parameters:** Learning rates and momentum values can impact the efficacy of the algorithm. A discussion on how to set them in practice is provided in Appendix D.

## 4  Synthetic Experiments

We validate that our algorithm recovers optimal low and high dimensional meta-parameters in two synthetic MNIST experiments with *Full FT Access*. Further details and results are provided in Appendix E, including a study of how our method performs comparably to differentiating exactly through the entire learning process of PT & FT, without approximations.

First, we optimize low dimensional meta-parameters characterizing a data augmentation scheme. We tune a 1-D meta-parameter $\phi$ representing the mean of a Normal distribution $\mathcal{N}(\phi, 1^2)$ from which we sample rotation augmentations to apply to PT images. FT images undergo rotations from a Normal distribution $\mathcal{N}(\mu_{\mathrm{FT}}, 1^2)$ with $\mu_{\mathrm{FT}} = 90°$; we therefore expect that $\phi$ should converge to near $\mu_{\mathrm{FT}}$. Using Algorithm 1 to optimize $\phi$ we find that the mean error in the optimized meta-parameter over 10 different initializations is small: $7.2 \pm 1.5°$, indicating efficacy of the algorithm.

Next, we consider learning high dimensional meta-parameters that characterize a PT per-example weighting scheme. The PT dataset contains some examples that have noisy labels, and FT examples all have clean labels. The meta-parameters are the parameters of a neural network that assigns importance weights to each PT example, which is used to weight the loss on that example during PT. We use Algorithm 1 again to optimize $\phi$, over 10 random initializations, finding the ratio of assigned importance weights between clean label PT examples and noisy label PT examples is greater than $10^2$. This is expected since the noisy label classes may worsen the quality of the PT model and so should be down-weighted.

## 5  Meta-Parameterized Multitask Pre-Training for Graph Neural Networks

We consider optimizing PT task weights for a multitask PT & FT problem of predicting the presence of protein functions (multitask binary classification) given graph-structured biological data as input. We have two experimental goals: first, in the *Full FT Access* setting, where methods are given access to all FT data at PT time, we evaluate whether optimizing task weighting meta-parameters can improve predictive performance on the FT tasks. Second, motivated by how in typical transfer learning problems, new tasks or labels not available at PT time may become available at FT time, we study the *Partial FT Access* setting, investigating how our method performs when it only sees *limited* FT tasks at PT time. In both settings, our method outperforms baselines.

### 5.1  Problem Setup

**Dataset and Task.** We consider the transfer learning benchmark introduced in [28], where the prediction problem at both PT and FT is multitask binary classification: predicting the presence/absence of specific protein functions ($y$) given a Protein-Protein Interaction (PPI) network as input (rep-

resented as a graph $\boldsymbol{x}$). The PT dataset has pairs $\mathcal{D}_{\mathrm{PT}} = \{(\boldsymbol{x}_i, y_i)\}_{i=1}^{|\mathcal{D}_{\mathrm{PT}}|}$, where $y \in \{0, 1\}^{5000}$ characterizes the presence/absence of 5000 particular protein functions. The FT dataset has pairs $\mathcal{D}_{\mathrm{FT}} = \{(\boldsymbol{x}_i, y_i)\}_{i=1}^{|\mathcal{D}_{\mathrm{FT}}|}$, where $y \in \{0, 1\}^{40}$ now characterizes the presence/absence of 40 different protein functions. Further dataset details in Appendix F.

**Meta-Parameterized Multitask PT.** To define a meta-parameterized PT scheme, we let meta-parameters $\phi \in \mathbb{R}^{5000}$ be weights for the binary PT tasks. Then, we define a PT loss incorporating the weights: $\mathcal{L}_{\mathrm{PT}} = \frac{1}{5000} \sum_{i=1}^{5000} 2\, \sigma(\phi_i)\, \mathcal{L}_{\mathrm{CE}}(f_{\mathrm{PT}}(\boldsymbol{x}; \boldsymbol{\theta}_{\mathrm{PT}}, \boldsymbol{\psi}_{\mathrm{PT}})_i, y_i)$, with $i$ indexing the tasks, $\sigma(\cdot)$ representing the sigmoid function (to ensure non-negativity and clamp the range of the weights), and $\mathcal{L}_{\mathrm{CE}}$ denoting the binary cross-entropy loss. With this loss defined, we use Algorithm 1 (with $P = 10$ PT steps and $K = 1$ truncated FT steps) to jointly learn $\phi$ and the feature extractor parameters $\boldsymbol{\theta}_{\mathrm{PT}}$. For computational efficiency, we only update the FT head when computing the FT best response Jacobian and keep the feature extractor of the model fixed. We use the training and validation splits of the FT dataset $\mathcal{D}_{\mathrm{FT}}$ proposed by the dataset creators [28] for computing the relevant gradient terms.

**Baselines.** Motivated by our goals, we compare with the following PT baselines:

- **No PT:** Do not perform PT (i.e., feature extractor parameters are randomly initialized).
- **Graph Supervised PT:** As explored in prior work on this domain [28], perform multitask supervised PT with $\mathcal{D}_{\mathrm{PT}}$. This corresponds to setting all task weights to 1: $\phi_i = 1, i = 1, \ldots, 5000$.
- **CoTrain:** A common baseline that makes use of the FT data available during PT [70] (like meta-parameterized PT). We PT a model with $5000 + 40$ outputs (covering the space of PT and FT labels) jointly on both $\mathcal{D}_{\mathrm{PT}}$ and $\mathcal{D}_{\mathrm{FT}}$. We do so by alternating gradient updates on batches sampled from each dataset in turn. Further details are in Appendix F.
- **CoTrain + PCGrad:** An extension of CoTrain, where we leverage the method PCGrad [72] to perform gradient projection and prevent destructive gradient interference between updates from $\mathcal{D}_{\mathrm{PT}}$ and $\mathcal{D}_{\mathrm{FT}}$. Further details and variants we tried are in Appendix F.

**Experimental Details.** We use a standardized setup to facilitate comparisons. Following [28], all methods use the Graph Isomorphism Network architecture [69], undergo PT for 100 epochs, and FT for 50 epochs, over 5 random seeds, using early stopping based on validation set performance. During FT, we initialize a new FT network head and either FT the whole network or freeze the PT feature extractor and learn the FT head alone (Linear Evaluation [50]). We report results for the strategy that performed best (full results in the appendix). We consider two experimental scenarios: (1) *Full FT Access:* Provide methods full access to $\mathcal{D}_{\mathrm{PT}}$ and $\mathcal{D}_{\mathrm{FT}}$ at PT time ($\mathcal{D}_{\mathrm{FT}}^{(\mathrm{Meta})} = \mathcal{D}_{\mathrm{FT}}$) and evaluate on the full set of 40 FT tasks; (2) *Partial FT Access:* Limit the number of FT tasks seen at PT time, by letting $\mathcal{D}_{\mathrm{FT}}^{(\mathrm{Meta})}$ include only 30 of the 40 FT tasks. At FT time, models are fine-tuned on the held-out 10 tasks not in $\mathcal{D}_{\mathrm{FT}}^{(\mathrm{Meta})}$. We use a 4-fold approach where we leave out 10 of the 40 FT tasks in turn, and examine performance across these 10 held-out tasks, over the folds.

## 5.2 Results

**Key Findings.** By optimizing PT task weights, meta-parameterized multitask PT improves performance on the FT problem of predicting presence/absence of protein functions given a protein-protein interaction graph as input. Performance improvements are also seen when generalizing to new FT tasks (protein functions), unseen at meta-PT time.

Table 1 presents quantitative results for the two experimental settings described. For the No PT and Graph Supervised PT baselines, we re-implement the methods from [28], obtaining improved results (full comparison in Appendix Table 5). In both full and partial FT access settings, meta-parameterized PT improves significantly on other methods, indicating that optimizing meta-parameters can improve predictive performance generally, and be effective even when new, related tasks are considered at evaluation time. Interestingly, we observe that CoTrain and CoTrain + PCGrad obtain relatively poor performance compared to other baselines; this could be because the methods overfit to the FT data during PT. Further analysis of this is presented in Appendix F.

**Further experiments.** In Appendix F, we study another partial FT access scenario with smaller $\mathcal{D}_{\mathrm{FT}}^{(\mathrm{Meta})}$, setting $\left|\mathcal{D}_{\mathrm{FT}}^{(\mathrm{Meta})}\right| = 0.5\,|\mathcal{D}_{\mathrm{FT}}|$, and find that meta-parameterized PT again outperforms other methods. (Table 7). We also examine another meta-parameter learning baseline, namely a version of CoTrain where we optimize task weights using a traditional hyperparameter optimization algorithm [42] jointly with the main model. We find that our method outperforms this baseline also (Table 5).

| Method | AUC ($\mathcal{D}_{\text{FT}}^{(\text{Meta})} = \mathcal{D}_{\text{FT}}$) | AUC ($\mathcal{D}_{\text{FT}}^{(\text{Meta})}$ excludes tasks) |
|---|---|---|
| No PT | $66.6 \pm 0.7$ | $65.8 \pm 2.5$ |
| Graph Supervised PT | $74.7 \pm 0.1$ | $74.8 \pm 1.8$ |
| CoTrain | $70.2 \pm 0.3$ | $69.3 \pm 1.8$ |
| CoTrain + PCGrad | $69.4 \pm 0.2$ | $68.1 \pm 2.3$ |
| Meta-Parameterized PT | $\mathbf{78.6 \pm 0.1}$ | $\mathbf{77.0 \pm 1.3}$ |

Table (1) **Meta-Parameterized PT improves predictive performance over baselines.** Table showing mean AUC and standard error for two evaluation settings. When provided all FT data at PT time (first results column), meta-parameterized PT significantly improves predictive performance. In a more challenging setting when $\mathcal{D}_{\text{FT}}^{(\text{Meta})}$ excludes FT tasks (10 of the 40 available tasks are held-out), evaluating mean AUC/standard error across four folds with each set of 10 FT tasks held out in turn, meta-parameterized PT again obtains the best performance: it is effective even with partial information about the downstream FT tasks.

**Analysis of learned structures.** In Appendix F, we conduct further analysis and study the effect of various PT strategies on the pre-trained representations (Figure 3), finding intuitive patterns of similarity between different methods. We also examine the learned task weights (Figure 4), and examine performance on a per-FT task basis with/without meta-parameterized PT (Figure 5), finding little evidence of negative transfer.

# 6 Meta-Parameterized SimCLR for Semi-Supervised Learning with ECGs

We now explore a second real-world application of our method: optimizing a data augmentation policy for self-supervised PT with SimCLR [8, 9] on electrocardiograms (ECGs). SimCLR is a popular self-supervised PT method that leverages data augmentations to define a contrastive PT objective (details in Appendix G.1). The choice/strength of the augmentations used significantly impacts the effectiveness of the algorithm [8]. In settings where relevant augmentations are known (e.g., natural images), SimCLR is readily applicable; however, for ECGs, effective augmentations are less clear, motivating the use of our algorithm to optimize the augmentation pipeline.

We have two experimental goals. Firstly, we examine the typical semi-supervised learning setting of *Full FT Access*: we explore whether optimizing the augmentations in SimCLR PT can improve performance on the supervised FT task of detecting pathologies from ECGs, given access to all FT data at meta-PT time. Secondly, to study the data efficiency of our method, we consider the *Partial FT Access* setting and explore performance given access to limited FT data at meta-PT time. We find that our method improves the performance of SimCLR, and that it is effective even with very limited amounts of FT data provided at meta-PT time.

## 6.1 Problem Setup

**Dataset and Task.** We construct a semi-supervised learning (SSL) problem using PTB-XL [64, 20], an open-source dataset of electrocardiogram (ECG) data. Let the model input at both PT and FT time be denoted by $x$, which represents a 12-lead (or channel) ECG sampled at 100 Hz for 10 seconds resulting in a $1000 \times 12$ signal. Our goal is to pre-train a model $f_{\text{PT}}$ on an unlabeled PT dataset of ECGs $\mathcal{D}_{\text{PT}} = \{x_i\}_{i=1}^{|\mathcal{D}_{\text{PT}}|}$ using SimCLR PT [8], and then fine-tune it on the labeled FT dataset $\mathcal{D}_{\text{FT}} = \{(x_i, y_i)\}_{i=1}^{|\mathcal{D}_{\text{FT}}|}$, where the FT labels $y \in \{0, 1\}^5$ encode whether the signal contains certain features indicative of particular diseases/pathologies. Further dataset details in Appendix G.

**ECG Data Augmentations.** To augment each ECG for SimCLR (example in Appendix G, Figure 6), we apply three transformations in turn (based on prior work in time series augmentation [30, 66]):

1. **Random cropping:** A randomly selected portion of the signal is zeroed out.
2. **Random jittering:** IID Gaussian noise is added to the signal.
3. **Random temporal warping:** The signal is warped with a random, diffeomorphic temporal transformation. This is formed by sampling from a zero mean, fixed variance Gaussian at each temporal location in the signal to obtain a velocity field, and then integrating and smoothing (following [4, 5]) to generate a temporal displacement field, which is applied to the signal.

|  | Test AUC at different FT dataset sizes $|\mathcal{D}_{\text{FT}}|$ | | | | |
|---|---|---|---|---|---|
| FT dataset size $|\mathcal{D}_{\text{FT}}|$ | 100 | 250 | 500 | 1000 | 2500 |
| No PT | $71.5 \pm 0.7$ | $76.1 \pm 0.3$ | $78.7 \pm 0.3$ | $82.0 \pm 0.2$ | $84.5 \pm 0.2$ |
| SimCLR | $74.6 \pm 0.4$ | $76.5 \pm 0.3$ | $79.8 \pm 0.3$ | $82.2 \pm 0.3$ | $85.8 \pm 0.1$ |
| Meta-Parameterized SimCLR | $\mathbf{76.1 \pm 0.5}$ | $\mathbf{77.8 \pm 0.4}$ | $\mathbf{81.7 \pm 0.2}$ | $\mathbf{84.0 \pm 0.3}$ | $\mathbf{86.7 \pm 0.1}$ |

Table (2)  **Meta-Parameterized SimCLR obtains improved semi-supervised learning performance.** Table showing mean AUC/standard error over seeds across 5 FT binary classification tasks for baselines and meta-parameterized SimCLR at different sizes of $\mathcal{D}_{\text{FT}}$, with $\mathcal{D}_{\text{FT}}^{(\text{Meta})} = \mathcal{D}_{\text{FT}}$. We observe improvements in performance with meta-parameterized SimCLR, which optimizes the augmentation pipeline.

**Meta-Parameterized SimCLR.** To construct a meta-parameterized SimCLR PT scheme, we instantiate meta-parameters $\phi$ as the weights of a neural network $w(\boldsymbol{x}; \phi)$ that takes in an input signal and outputs the warp strength: the variance of the Gaussian that is used to obtain the velocity field for temporal warping. This parameterization permits signals to be warped more/less aggressively depending on their individual structure. With this definition, the SimCLR PT loss is directly a function of the meta-parameters, and we can use Algorithm 1 (with $P = 10$ PT steps and $K = 1$ truncated FT steps) to jointly learn $\phi$ and the feature extractor parameters $\boldsymbol{\theta}_{\text{PT}}$. For computational efficiency, we only update the FT head when computing the FT best response Jacobian and keep the feature extractor of the model fixed. We use the training and validation splits of the FT dataset $\mathcal{D}_{\text{FT}}$ proposed by the dataset creators [64] for computing the relevant gradient terms.

**Baselines.** Our experimental goals suggest the following PT baselines:

- **No PT:** Do not perform PT (i.e., feature extractor parameters are randomly initialized).
- **SimCLR:** Pre-train a model using SimCLR with the above three augmentations *without* learning per-example temporal warping strengths.

**Experimental Details.** We standardize the experimental setup to facilitate comparisons. All methods use a 1D CNN based on a ResNet-18 [23] architecture. The temporal warping network $w(\boldsymbol{x}; \phi)$ is a four layer 1D CNN. SimCLR PT takes place for 50 epochs for all methods, over three PT seeds. At evaluation time, for all methods, we initialize a new FT network head over the PT network feature extractor and FT the whole network for 200 epochs, over five FT seeds. Validation set AUC is used for early stopping. We consider two experimental settings: (1) *Full FT Access*, standard SSL: consider different sizes of the labelled FT dataset $\mathcal{D}_{\text{FT}}$ and make all the FT data available at meta-PT time, $\mathcal{D}_{\text{FT}}^{(\text{Meta})} = \mathcal{D}_{\text{FT}}$; and (2) *Partial FT Access*, examining data efficiency of our algorithm: SSL when only limited FT data is available at meta-PT time: $\mathcal{D}_{\text{FT}}^{(\text{Meta})} \subseteq \mathcal{D}_{\text{FT}}$. We evaluate performance across the 5 binary classification tasks in both settings. Further details are provided in Appendix G.

### 6.2   Results

**Key Findings.** By optimizing the data augmentation policy used in SimCLR PT, meta-parameterized SimCLR improves performance on the FT problem of detecting pathologies from ECG data. Even a small amount of FT data provided at meta-PT time can lead to improved FT performance.

Table 2 shows results for the *Full FT Access* setting, $\mathcal{D}_{\text{FT}}^{(\text{Meta})} = \mathcal{D}_{\text{FT}}$: mean AUC/standard error over seeds across the 5 FT binary classification tasks at different sizes of $\mathcal{D}_{\text{FT}}$. We observe that meta-parameterized SimCLR improves on other baselines in all settings. Note that while these gains are modest, they are obtained with simple augmentation policies; our method may yield further improvements if applied to policies with more scope to specialize the augmentations.

Next, we consider the *Partial FT Access* scenario where $\mathcal{D}_{\text{FT}}^{(\text{Meta})} \subseteq \mathcal{D}_{\text{FT}}$, which is relevant when we only have a small amount of FT data at meta-PT time. Fixing $|\mathcal{D}_{\text{FT}}| = 500$, we find that with $|\mathcal{D}_{\text{FT}}^{(\text{Meta})}|$ as small as 50, we obtain test AUC of $81.3 \pm 0.5$, compared to $79.8 \pm 0.3$ with no optimization of augmentations: this shows that even small $|\mathcal{D}_{\text{FT}}^{(\text{Meta})}|$ appear to be sufficient for meta-parameter learning. Further results showing performance curves varying $|\mathcal{D}_{\text{FT}}^{(\text{Meta})}|$ are in Appendix G.

**Further experiments.** In Appendix G, we study other aspects of our method on this domain, including: (1) Exploring different values of $K$, the number of FT steps differentiated through when obtaining meta-parameter gradients; and (2) Examining a meta-parameter learning baseline where

augmentations are optimized for supervised learning, using the method in [42], and then applied to semi-supervised learning (to compare how optimizing augmentations for supervised learning compares to optimizing them for semi-supervised learning). We find that our method is not very sensitive to the value of $K$ (provided $K > 0$), and that it outperforms this additional baseline.

## 7 Related Work

**Gradient-based hyperparameter optimization (HO):** Gradient-based HO roughly falls into two camps. The simpler and less scalable approach differentiates through training [12, 44]. The other approach assumes that optimization reaches a fixed point, and approximates the best-response Jacobian [7, 41, 43, 42]. Neither of these approaches can be straightforwardly applied to scalably differentiate through two stages of optimization (PT & FT). Direct differentiation through both stages would be too memory-intensive. Approximating the best-response Jacobian using the IFT as in [42] twice is feasible, but requires changing the FT objective to include a proximal term [55], and tuning two sets of interacting approximations. Instead, we compose a constant-memory IFT approximation for the lengthy PT stage with an exact backprop-through-training for the shorter FT stage.

**Applications of Nested Optimization:** Many prior works frame learning as nested optimization, including few-shot learning [16, 1, 17, 55, 21, 58, 53, 75, 31, 38], neural network teaching [14, 15, 62, 54], learning data augmentation and reweighting strategies [32, 22, 57, 60, 29], and auxiliary task learning [49, 51, 39]. The majority of this work studies nested optimization in the standard one-stage supervised learning paradigm, unlike our setting: the two-stage PT & FT problem. The most closely related works to ours are [70], where PT task weights are learned for a multitask PT problem using electronic health record data, and [71], where a masking policy is learned for masked language modelling PT. In contrast to our work, which introduces the more general framing of meta-parameter optimization, [70] and [71] are focused only on specific instantiations of meta-parameters as task weights and masking policies. The learning algorithms in these works either: differentiate directly through truncated PT & FT [71] (which may not be scalable to longer PT/large encoder models), or leverage extensive first-order approximations [70], unlike our more generally applicable approach.

## 8 Scope and Limitations

Our gradient-based algorithm applies in situations where we want to optimize (potentially high-dimensional) PT hyperparameters, or *meta-parameters*, and have access to a model, PT data, and FT data. We demonstrated that even limited FT data availability can be sufficient to guide meta-parameter learning; however, our method would not apply when no FT data at all is available at meta-PT time, or if the model or PT data were not available. Our algorithm requires meta-parameters to be differentiable, and cannot directly be used to optimize meta-parameters that do not affect the PT optimization landscape (e.g., PT learning rates).

## 9 Conclusion

In this work, we studied the problem of optimizing high-dimensional pre-training (PT) hyperparameters, or *meta-parameters*. We formalized *Meta-Parameterized Pre-Training*, a variant of standard PT incorporating these meta-parameters, and proposed a gradient-based algorithm to efficiently learn meta-parameters by approximately differentiating through the two-stage PT & FT learning process. In experiments, we used our algorithm to improve predictive performance on two real-world PT tasks: multitask PT with graph structured data [28], and self-supervised contrastive PT on electrocardiogram signals using SimCLR [8]. Future work could apply our method to learn other potential instantiations of meta-parameters, such as learned auxiliary tasks and noise models.

**Societal Impact.** Our contribution in this work is methodological, namely a new algorithm to optimize high-dimensional pre-training hyperparameters. We do not expect there to be direct negative societal impacts of this contribution. However, to evaluate our method, we considered an experimental domain using healthcare data. Given the high risk nature of this domain, before use in real-world settings, the method should be validated in retrospective and prospective studies. This is to detect any failure modes and identify potential harm that may come from deploying it.

## Acknowledgements

This work was supported in part by funds from Quanta Computer, Inc. The authors thank the members of the Clinical and Applied Machine Learning group at MIT and Paul Vicol for helpful feedback.

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
