# B   Notation and Acronyms

| | |
|---|---|
| PT | Pre-training |
| FT | Fine-tuning |
| AUROC (AUC) | Area Under Receiver-Operator Characteristic |
| $\bullet$ | Placeholder for either PT or FT |
| $\mathcal{X}$ | Input domain to models |
| $\boldsymbol{x}$ | Model input |
| $\mathcal{Y}_\bullet$ | True output space (e.g., space of labels) |
| $y$ | Label |
| $\hat{\mathcal{Y}}_\bullet$ | Prediction output space of a model |
| $\hat{y}$ | Predicted Label |
| $\Theta$ | Parameter space for feature extractors of models |
| $\boldsymbol{\theta}$ | Feature extractor parameters |
| $\Psi_\bullet$ | Parameter space for prediction head of model (output layer) |
| $\boldsymbol{\psi}_\bullet$ | Head parameters |
| $\Phi$ | Space of meta-parameters |
| $\phi$ | Meta-parameters |
| $f_\bullet : \mathcal{X}; \Theta, \Psi_\bullet \to \hat{\mathcal{Y}}_\bullet$ | General parameteric model satisfying compositional structure $f_\bullet = f_\bullet^{(\mathrm{head})} \circ f^{(\mathrm{feat})}$ |
| $f^{(\mathrm{feat})}(\cdot; \boldsymbol{\theta} \in \Theta)$ | Feature extractor that is transferable across learning stages (e.g., PT to FT) |
| $f_\bullet^{(\mathrm{head})}(\cdot; \boldsymbol{\psi} \in \Psi_\bullet)$ | Output 'head' of a model that is stage-specific and not transferable. |
| $\mathcal{D}_\bullet$ | General data distribution or dataset |
| $\mathcal{L}_\bullet : \hat{\mathcal{Y}}_\bullet \times \mathcal{Y}_\bullet \to \mathbb{R}$ | Loss function |
| $\mathcal{L}_{\mathrm{CE}}$ | Example loss function: cross-entropy |
| $L_\bullet(\boldsymbol{\theta}, \boldsymbol{\psi}_\bullet; \mathcal{D}_\bullet)$ | Expected loss over a data distribution $\mathbb{E}_{\mathcal{D}_\bullet}\left[\mathcal{L}_\bullet(f_\bullet(\boldsymbol{x}_\bullet; \boldsymbol{\theta}, \boldsymbol{\psi}_\bullet), y_\bullet)\right]$. |
| $\mathrm{Alg}_\bullet$ | Learning algorithm used for optimization (e.g., stochastic gradient descent) |
| $g(\phi)$ | Meta-parameter optimization objective $L_{\mathrm{FT}}\left(\mathrm{Alg}_{\mathrm{FT}}\left(\mathrm{Alg}_{\mathrm{PT}}(\boldsymbol{\theta}_{\mathrm{PT}}^{(0)}, \boldsymbol{\psi}_{\mathrm{PT}}^{(0)}; \mathcal{D}_{\mathrm{PT}}, \phi), \boldsymbol{\psi}_{\mathrm{FT}}^{(0)}; \mathcal{D}_{\mathrm{FT}}\right); \mathcal{D}_{\mathrm{FT}}\right)$ |
| $\phi^{(\mathrm{opt})}$ | Optimal meta-parameters satisfying $\phi^{(\mathrm{opt})} = \mathrm{argmin}_{\phi \in \Phi}\, g(\phi)$ |
| $\boldsymbol{\theta}_{\mathrm{PT}}^*(\phi)$ | PT best response values satisfying $\boldsymbol{\theta}_{\mathrm{PT}}^*(\phi) = \mathrm{Alg}_{\mathrm{PT}}(\boldsymbol{\theta}_{\mathrm{PT}}^{(0)}, \boldsymbol{\psi}_{\mathrm{PT}}^{(0)}; \mathcal{D}_{\mathrm{PT}}, \phi)$ |
| $\boldsymbol{\theta}_{\mathrm{FT}}^*(\phi), \psi_{\mathrm{FT}}^*(\phi)$ | FT best response values satisfying $\boldsymbol{\theta}_{\mathrm{FT}}^*(\phi), \psi_{\mathrm{FT}}^*(\phi) = \mathrm{Alg}_{\mathrm{FT}}(\boldsymbol{\theta}_{\mathrm{PT}}^*(\phi), \boldsymbol{\psi}_{\mathrm{FT}}^{(0)}; \mathcal{D}_{\mathrm{FT}})$ |
| $\frac{\partial g}{\partial \phi}$ | Gradient w.r.t. meta-parameters, which we compute for gradient-based optimization of $\phi$ |
| $\left[\boldsymbol{\theta}_{\mathrm{FT}}, \quad \boldsymbol{\psi}_{\mathrm{FT}}\right]$ | Shorthand to representation concatenation of parameter vectors. |
| $\frac{\partial L_{\mathrm{FT}}}{\partial\left[\boldsymbol{\theta}_{\mathrm{FT}}, \quad \boldsymbol{\psi}_{\mathrm{FT}}\right]}$ | FT loss gradient: first term in meta-parameter gradient. |
| $\frac{\partial \mathrm{Alg}_{\mathrm{FT}}}{\partial \boldsymbol{\theta}_{\mathrm{PT}}}$ | FT best response Jacobian: second term in meta-parameter gradient. |
| $\frac{\partial \mathrm{Alg}_{\mathrm{PT}}}{\partial \phi}$ | PT best response Jacobian: third term in meta-parameter gradient. |
| $K$ | Number of steps we unroll in FT to compute FT best response Jacobian. |
| $P$ | Number of PT steps before each meta-parameter update. |
| $\mathrm{copy}(\theta)$ | Make a copy of the parameters $\theta$ such that gradients do not flow through (like a stop-gradient). |
| $\mathcal{D}_{\mathrm{FT}}^{(\mathrm{tr})}$ | Training split of the FT data set, used during meta-parameter learning for updating the FT parameters. |
| $\mathcal{D}_{\mathrm{FT}}^{(\mathrm{val})}$ | Validation split of the FT data set, used during meta-parameter learning for optimizing meta-parameters. |
| $\mathcal{D}_{\mathrm{FT}}^{(\mathrm{Meta})}$ | FT data available at PT time for meta-parameter learning. We have that $\mathcal{D}_{\mathrm{FT}}^{(\mathrm{Meta})} = \mathcal{D}_{\mathrm{FT}}^{(\mathrm{tr})} \cup \mathcal{D}_{\mathrm{FT}}^{(\mathrm{val})} \subseteq \mathcal{D}_{\mathrm{FT}}^{(\mathrm{all})}$. |
| IFT | Implicit Function Theorem |
| GIN | Graph Isomorphism Network |
| ECG | Electrocardiogram |
| $\eta_{\mathrm{PT}}$ | learning rate for PT |
| $\eta_{\mathrm{FT}}$ | learning rate for FT |
| $\eta_{\mathrm{V}}$ | learning rate for meta parameters |

Table (3)   Notation

# C  Our Algorithm: Further Details

---

**Algorithm 2** Gradient-based algorithm to learn meta-parameters, incorporating other practical details not present in the main paper description. Notation defined in Table 3. Note that vector-Jacobian products (VJPs) can be efficiently computed by standard autodifferentiation.

---

1: Initialize PT parameters $\boldsymbol{\theta}_{\text{PT}}^{(\text{init})}, \boldsymbol{\psi}_{\text{PT}}^{(\text{init})}, \boldsymbol{\psi}_{\text{FT}}^{(\text{init})}$ and meta-parameters $\boldsymbol{\phi}^{(0)}$
2: **for** $n = 1, \ldots, N$ iterations **do**
3:     Initialize $\boldsymbol{\theta}_{\text{PT}}^{(0)} = \boldsymbol{\theta}_{\text{PT}}^{(\text{init})}$ and $\boldsymbol{\psi}_{\text{PT}}^{(0)} = \boldsymbol{\psi}_{\text{PT}}^{(\text{init})}$.
4:     **for** $p = 1, \ldots, P$ PT iterations **do**
5:         $\left[\boldsymbol{\theta}_{\text{PT}}^{(p)}, \boldsymbol{\psi}_{\text{PT}}^{(p)}\right] = \left[\boldsymbol{\theta}_{\text{PT}}^{(p-1)}, \boldsymbol{\psi}_{\text{PT}}^{(p-1)}\right] - \eta_{\text{PT}} \left.\frac{\partial L_{\text{PT}}}{\partial[\boldsymbol{\theta}_{\text{PT}}, \boldsymbol{\psi}_{\text{PT}}]}\right|_{\boldsymbol{\theta}_{\text{PT}}^{(p-1)}, \boldsymbol{\psi}_{\text{PT}}^{(p-1)}}$     `# Unrolled step of` $\text{Alg}_{\text{PT}}$
6:     **end for**
7:     **if** $n < N_{\text{warmup}}$ **then**
8:         Update PT initialization by setting: $\boldsymbol{\theta}_{\text{PT}}^{(\text{init})} = \boldsymbol{\theta}_{\text{PT}}^{(P)}$ and $\boldsymbol{\psi}_{\text{PT}}^{(\text{init})} = \boldsymbol{\psi}_{\text{PT}}^{(P)}$
9:         Skip meta-parameter update and continue
10:    **end if**
11:    Initialize FT parameters $\boldsymbol{\psi}_{\text{FT}}^{(0)} = \boldsymbol{\psi}_{\text{FT}}^{(\text{init})}$ and $\boldsymbol{\theta}_{\text{FT}}^{(0)} = \texttt{copy}(\boldsymbol{\theta}_{\text{PT}}^{(P)})$.
12:    Approximate $\boldsymbol{\theta}_{\text{FT}}^*, \boldsymbol{\psi}_{\text{FT}}^*$ using (4), with $\mathcal{D}_{\text{FT}}^{(\text{tr})}$.
13:    Compute $g_1 = \left.\frac{\partial L_{\text{FT}}}{\partial[\boldsymbol{\theta}_{\text{FT}}, \quad \boldsymbol{\psi}_{\text{FT}}]}\right|_{\boldsymbol{\theta}_{\text{FT}}^*, \boldsymbol{\psi}_{\text{FT}}^*}$, using $\mathcal{D}_{\text{FT}}^{(\text{val})}$.   `# FT Loss gradient`
14:    Compute VJP $g_2 = g_1 \left.\frac{\partial \text{Alg}_{\text{FT}}}{\partial \boldsymbol{\theta}_{\text{PT}}}\right|_{\boldsymbol{\theta}_{\text{PT}}^{(P)}, \boldsymbol{\psi}_{\text{FT}}^{(0)}}$ using the unrolled learning step from line 12, and $\mathcal{D}_{\text{FT}}^{(\text{tr})}$.
15:    Approximate VJP $\left.\frac{\partial g}{\partial \boldsymbol{\phi}}\right|_{\boldsymbol{\phi}^{(n-1)}} = g_2 \left.\frac{\partial \text{Alg}_{\text{PT}}}{\partial \boldsymbol{\phi}}\right|_{\boldsymbol{\phi}^{(n-1)}}$ using IFT (3).
16:    $\boldsymbol{\phi}^{(n)} = \boldsymbol{\phi}^{(n-1)} - \eta_{\text{V}} \left.\frac{\partial g}{\partial \boldsymbol{\phi}}\right|_{\boldsymbol{\phi}^{(n-1)}}$     `# Update meta-parameters`
17:    Update PT initialization by setting: $\boldsymbol{\theta}_{\text{PT}}^{(\text{init})} = \boldsymbol{\theta}_{\text{PT}}^{(P)}$ and $\boldsymbol{\psi}_{\text{PT}}^{(\text{init})} = \boldsymbol{\psi}_{\text{PT}}^{(P)}$.
18:    Update FT initialization by setting: $\boldsymbol{\psi}_{\text{FT}}^{(\text{init})} = \boldsymbol{\psi}_{\text{FT}}^*$.
19: **end for**

---

We include a more detailed algorithm in Algorithm 2 reflecting certain extra details that were excluded in the main text due to space restrictions. We discuss some of these details here.

**$\mathcal{D}_{\text{FT}}$ splits.**  In practice, we have access to finite datasets and use minibatches, rather than data generative processes. Following standard convention, we split $\mathcal{D}_{\text{FT}}$ into two subsets for meta-learning: $\mathcal{D}_{\text{FT}}^{(\text{tr})}$ and $\mathcal{D}_{\text{FT}}^{(\text{val})}$ (independent of any held-out $\mathcal{D}_{\text{FT}}$ testing split), and define the FT data available at meta-PT time as $\mathcal{D}_{\text{FT}}^{(\text{Meta})} = \mathcal{D}_{\text{FT}}^{(\text{tr})} \cup \mathcal{D}_{\text{FT}}^{(\text{val})}$. We use $\mathcal{D}_{\text{FT}}^{(\text{tr})}$ for the computation of $\left.\frac{\partial \text{Alg}_{\text{FT}}}{\partial \boldsymbol{\theta}_{\text{PT}}}\right|_{\boldsymbol{\theta}_{\text{PT}}^{(P)}, \boldsymbol{\psi}_{\text{FT}}^{(0)}}$ and $\left.\frac{\partial \text{Alg}_{\text{PT}}}{\partial \boldsymbol{\phi}}\right|_{\boldsymbol{\phi}^{(n-1)}}$ and $\mathcal{D}_{\text{FT}}^{(\text{val})}$ for the computation of $\left.\frac{\partial L_{\text{FT}}}{\partial[\boldsymbol{\theta}_{\text{FT}}, \quad \boldsymbol{\psi}_{\text{FT}}]}\right|_{\boldsymbol{\theta}_{\text{FT}}^*, \boldsymbol{\psi}_{\text{FT}}^*}$ in Algorithm 2. The description in Algorithm 2 includes details of the different datasets used for different computations.

**Online updates.**  Given that PT phases often involve long optimization horizons, we update $\boldsymbol{\theta}_{\text{PT}}$ and $\boldsymbol{\psi}_{\text{PT}}$ online, jointly with $\boldsymbol{\phi}$, rather than re-initializing them at every meta-iteration (see Algorithm 2).

FT phases are typically shorter so we could in theory re-initialize $\boldsymbol{\psi}_{\text{FT}}$ at each meta-iteration, as is presented in the main text, Algorithm 1. However, for further computational and memory efficiency, in our experiments, we also optimize these parameters online. For $\boldsymbol{\psi}_{\text{FT}}$ this makes each meta-iteration resemble a "warm-start" to the FT problem. In the updated description in Algorithm 2, we update the notation for the FT head to describe this.

See below for a discussion on optimization horizons and considerations when jointly optimizing meta-parameters with PT and FT parameters.

**Notational clarification: vector concatenation.** We use the notation: $[\boldsymbol{\theta}_{\mathrm{FT}}, \quad \boldsymbol{\psi}_{\mathrm{FT}}]$ to represent concatenation of the two vectors $\boldsymbol{\theta}_{\mathrm{FT}}$ and $\boldsymbol{\psi}_{\mathrm{FT}}$. The output of $\mathrm{Alg}_{\mathrm{FT}}$ contains two parameter vectors, and these are implicitly concatenated to make sure that dimensionalities agree in the algorithm.

**Warmup iterations.** We can optionally include *warmup iterations* where we optimize the PT parameters and do not perform updates to the meta-parameters. This is to ensure that the PT parameters are a reasonable approximation of $\mathcal{L}_{\mathrm{PT}}$-optimal parameters. The description in the algorithm is updated to reflect this, with the $N_{\mathrm{warmup}}$ reflecting the number of warmup iterations.

**On optimization horizons.** Prior work [67] has suggested that online optimization of certain hyperparameters (such as learning rates) using short horizons may yield suboptimal solutions. This is known as the short-horizon bias (SHB) problem. We now discuss this concern further in the context of our algorithm.

- **What is the short-horizon bias (SHB) problem?** SHB is understood to be a special case of the bias induced by truncating telescoping sums for optimization parameters. The effects of the truncation can be pronounced with optimization parameters [67], but there exist methods like [6] to deal with these if they occur.

- **Do we expect this to be a concern in our setting?** There are two hypergradients in our system that could suffer from bias: the PT hypergradients and the FT hypergradients. In both cases, the impact from biased hypergradients appears to be minimal. We will argue for this claim through each hypergradient term separately.

  **PT Hypergradient:** The PT hypergradient does not suffer from the short-horizon bias because the PT model is expected to have approximately converged at each hyperparameter update. This is not only a requirement of the implicit function theorem and the algorithm from [42] to apply, but also is directly enforced in our system through the use of online-updates and a warmup period (see Algorithm 2).

  **FT Hypergradient:** For the gradient through FT, we acknowledge that differentiating through only one step could, in theory, produce biased hypergradients. However, several prior works on meta-learning various structures similar to what we consider [51, 54, 42, 49, 29] did not observe significant bias. Therefore, from an empirical standpoint, this bias is not necessarily expected to be a significant issue.

  As seen in our experimental results, we also observe improved experimental results by setting $K = 1$ in our algorithm, suggesting minimal SHB impact. To study this issue further, we include experiments comparing to full backpropagation through PT and FT in synthetic experiments (Appendix E), and compare different values of $K$ in our semi-supervised learning experiments (Appendix G).

- **Why might SHB not be a concern with the hyperparameters we consider?** As stated, the SHB issue has mainly been observed in the context of optimization hyperparameters such as the learning rate. This could be because the learning rate directly affects the rate at which we approach the critical point, but it does not directly change the critical point. As seen in the analysis in [67], optimizing the LR with short rollouts results in (far too aggressively) decaying the step size to decrease variance and converge faster. In contrast, other hyperparameters, like weight decay or augmentations, directly change the fixed point that we are converging to (as opposed to just the rate). In setups where the hyperparameters directly affect the fixed point, SHB has not been observed — for example, see [57, 43]. These works do online, limited horizon optimization of hyperparameters directly affecting the critical point.

# D  Practical Heuristics for Tuning Optimizer Parameters

The optimization parameters used in nested optimization can be crucial for success. In our synthetic experiments in Section 4 we were able to use default optimizer selections; however these settings may not work in practice for all domains (as seen in our real-world experiments). Here, we list some basic guidelines a practitioner can iterate through to debug meta-parameter optimization.

**Step 1: What to do if the meta-parameters are changing wildly?**  First, decrease the learning rate for the meta-parameters. Momentum parameters can be dangerous – see [18]; using an optimizer without momentum may work better in some situations. If the meta-parameters begin oscillating later into training, try decreasing momentum.

**Step 2: What to do if the pre-training parameters are changing wildly?**  First, make sure the meta-parameters are not moving around rapidly. Once the meta-parameters are stable, you should be able to decrease the learning rate of the pre-training optimizer until convergence.

**Step 3: What to do if the meta-parameters are not changing?**  First, make sure that your pre-training parameters are finding good solutions by examining the pre-training optimization and optimizer settings. Next, make sure that the IFT is giving a good approximation for the pre-training response. You should begin with 1 Neumann term (or an identity inverse-Hessian approximation), because this often works well; see [42]. If 1 Neumann term works, you can try adding more until they offer no benefit. Next, make sure that differentiation through optimization is giving a reasonable gradient. If the unrolled optimizer is diverging, this will not give us useful gradients, so we must make sure these FT parameters converge. After you verify these components, try increasing the meta-parameter learning rate.

# E    Synthetic Experiments: Further Details

We discuss further details on the synthetic experiments. All experiments in this section were run on Google Colab, using the default GPU backend.

## E.1    Meta-Parameterized Data Augmentation

Here, we have additional details for the data augmentation synthetic experiments in Section 4.

**Dataset.**    Both PT and FT tasks are supervised MNIST digit classification (i.e., a 10-class classification problem). Our pre-training dataset is 3000 randomly-sampled MNIST data points. The fine-tuning training and validation sets are a distinct set of 3000 randomly-sampled MNIST data points augmented with a rotational degree drawn from $\mathcal{N}(\mu, \sigma^2)$ for some mean $\mu$ and standard deviation $\sigma$.

In the main text (Section 4) we studied the situation where the FT rotation distribution was $\mathcal{N}(\mu, 1^2)$, $\mu = 90°$; note that the standard deviation is fixed at 1. We also examine here a situation where we try to learn both the mean and standard deviation (results in Figure 2a), where the FT rotation distribution is $\mathcal{N}(45, 15^2)$.

**Defining meta-parameters.**    Our meta-parameters parameterize a rotational augmentation distribution that we apply to the PT images, $\mathcal{N}(\mu_{\mathrm{PT}}, \sigma_{\mathrm{PT}}^2)$. We consider two scenarios. First, where we only optimize the mean: $\phi = \{\mu_{\mathrm{PT}}\}$, and $\sigma_{\mathrm{PT}}$ is fixed to 1, which is the situation in Section 4. In this case, the initialization of $\phi$ is sampled uniformly from $[45, 135]$. Second, we optimize both the mean and the standard deviation of the rotation distribution: $\phi = \{\mu_{\mathrm{PT}}, \sigma_{\mathrm{PT}}\}$ (results in Figure 2a). In both settings, we expect the optimal PT rotation distribution for augmentations to be equal to what is used at FT time.

**Model architectures.**    Our model is a fully-connected feedforward network, with 1 hidden layer with 64 hidden units and a ReLU activation.

**Algorithm and Implementation details.**    We are able to use implicit differentiation with 1 Neumann term for the pre-training, and 1 step of differentiation through optimization for the fine-tuning training step. We use an Adam optimizer with a LR of 0.01 for pre-training and 0.3 for the meta-parameters. For fine-tuning, we use SGD (to match exactly the methods description in (4)) with the default learning rate of 0.01. We train with a batch size of 64 for each optimizer for 5 epochs. We alternate between taking 1 step of optimization for each set of parameters: $N_{\mathrm{warmup}} = 0, K = 1, P = 1$. These hyperparameters were chosen based on a simple strategy discussed in Section D, without particular tuning.

**Experimental setup.**    For the main experiments where we learn only the mean, we consider 10 different sampled mean initializations from $[45, 135]$. For the additional experiments where we learn the mean and the standard deviation, we fix the target distribution at $\mathcal{N}(45, 15^2)$ and examine two initializations: $\phi^{(0)} = \{0, 1\}$ and $\phi^{(0)} = \{90, 1\}$.

**Results.**    When learning the mean, we are able to approximately recover the true rotation distribution after training with a final difference mean and standard error of $7.2 \pm 1.5°$, over 10 sampled mean rotations, indicating efficacy of the algorithm.

Next, we examine the results for learning the mean and standard deviation from different initializations, in Figure 2a, and observe that we can approximately recover the true augmentation distribution from both initializations.

## E.2    Meta-Parameterized Per-Example Weighting

Here, we have additional details for the example weighting synthetic experiments in Section 4.

**Dataset.**    PT and FT tasks are again based on supervised MNIST image classification. The PT task is adjusted to be a 1000-class problem, where MNIST digits in classes 0-4 keep their original labels,

and MNIST digits in classes 5-9 are now assigned a noisy label: a random label between 5 and 1000. Our PT set is 3000 randomly-sampled MNIST data points. We use the standard MNIST training set for pre-training, and if any data point is in class 5-9 we noise it, by assigning a random label between 5 and 1000. We use the standard MNIST testing set for FT, split into a FT training and validation set. The FT set contains only images with classes 0-4.

**Defining meta-parameters.** Our meta-parameters $\phi$ are the parameters of a *weighting CNN* that assigns an importance weight to each PT data point, which is then used to weight the loss on that data point during PT. We expect the optimal weighting strategy to assign maximal weight to PT images in classes 0-4, since these are not noisy and are seen at FT time, and minimal weight to the other images, since these have noisy labels.

**Model architectures.** Our weighting network has an architecture of two convolutional layers, then a fully-connected layer. The first layer has 32 filters with a kernel size of 5, followed by batch-norm, with a ReLU activation and max pooling. The second convolutional layer is the same as the first, except with 64 filters and a kernel size of 3. The fully-connected layer has a 1-dimensional output with an activation of $2\sigma$ applied, so the output is in $(0, 2)$.

**Implementation details.** As with the MNIST augmentation experiments, we are able to use implicit differentiation with 1 Neumann terms for the PT, and 1 step of differentiation through optimization for the FT training. Again, we use an Adam optimizer with default parameters for PT and the meta-parameters, and SGD with default learning rate of 0.01 for FT. We use a batch-size of 100 for each optimizer and train each seed for 100 epochs. We alternate between taking 1 step of optimization for each set of parameters: $N_{\text{warmup}} = 0, K = 1, P = 1$. These hyperparameters were chosen based on a simple strategy discussed in Section D, without particular tuning.

**Results.** Using Algorithm 1 once again, we find that PT images from class 0-4 are assigned high weight, and those from classes 5-1000 are assigned low weight. This is an expected result: since the PT classes 0-4 are also the FT classes, we expect images from these classes to be upweighted. PT images not from these classes do not appear at FT and have noisy labels, hence are downweighted. This result is visualized in Figure 2b.

### E.3 The Impact of Approximating Meta-Parameter Gradients

We now study how using the two gradient approximations in our algorithm compare to storing the entire PT and FT process in GPU memory and differentiating through the whole process to obtain meta-parameter gradients. We consider our first synthetic setting, where we aim to learn rotation augmentations for MNIST PT, given that the FT set is augmented in a specific way. In the following experiments, the FT set is augmented with rotations drawn from $\mathcal{N}(90, 1)$.

**Experimental setup.** We compare the following methods to study the impact of the gradient approximations.

- Backpropagation through training (BPTT): The PT augmentation distribution is initialized to $\mathcal{N}(45, 1)$. We do 500 steps of PT and 500 steps of FT steps, and use BPTT (through these 1000 optimization steps) to optimize the augmentations. This is near the limit of what we could fit into our GPU memory. This process is then repeated for 500 hyperparameter optimization steps.

- Meta-parameterized PT: We run our algorithm. The PT augmentation distribution is initialized to $\mathcal{N}(45, 1)$. We set $P = 1$, $K = 1$, running for 500 PT and FT steps overall (for a fair comparison with BPTT).

- Optimal augmentations: We set the PT augmentation distribution to be the optimal setting (i.e., identical to that used for FT): $\mathcal{N}(90, 1)$. This is also run for 500 PT and 500 FT steps.

- Initialization augmentations: We set the PT augmentation distribution to be: $\mathcal{N}(45, 1)$ as a baseline. This is also run for 500 PT and 500 FT steps.

**Results.**

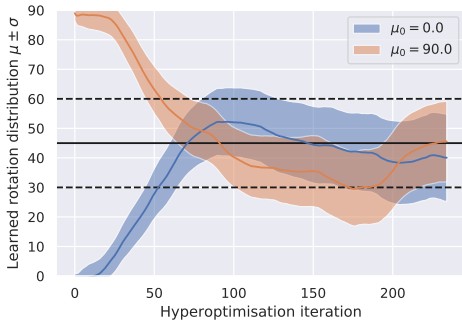

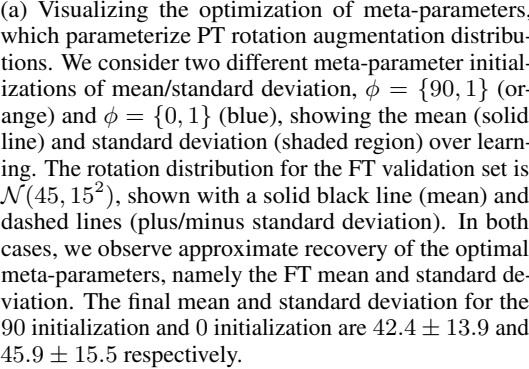

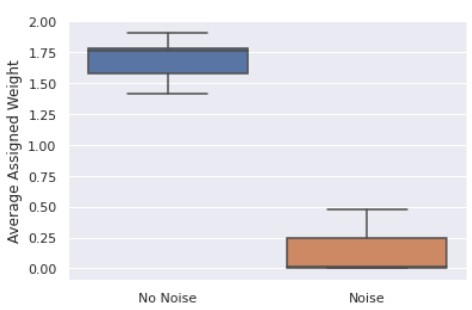

(a) Visualizing the optimization of meta-parameters, which parameterize PT rotation augmentation distributions. We consider two different meta-parameter initializations of mean/standard deviation, $\phi = \{90, 1\}$ (orange) and $\phi = \{0, 1\}$ (blue), showing the mean (solid line) and standard deviation (shaded region) over learning. The rotation distribution for the FT validation set is $\mathcal{N}(45, 15^2)$, shown with a solid black line (mean) and dashed lines (plus/minus standard deviation). In both cases, we observe approximate recovery of the optimal meta-parameters, namely the FT mean and standard deviation. The final mean and standard deviation for the 90 initialization and 0 initialization are $42.4 \pm 13.9$ and $45.9 \pm 15.5$ respectively.

(b) The distribution of importance weights assigned to examples with/without noisy labels, over 10 random seeds of weights, produced by a weighting CNN. We show the average weight applied to non-noised and noised examples, normalized by dividing by the sum of the data weights. The weighted CNN has recovered the desired solution of down-weighting examples with noisy labels, indicating successful learning of high-dimensional meta-parameters.

Figure (2)   **Results for learning pre-training augmentation meta-parameters.**

- BPTT without compute limitations: Running BPTT for 500 hyperparameter optimization steps takes about 20 hours. Doing so, it achieves a test accuracy of 88.0%.

- Meta-parameterized PT: Running our method takes about 30 minutes. This achieves a test accuracy of 87.6%.

- BPTT with compute limitations: Limiting the compute budget of BPTT to be similar to our method, it obtains a test accuracy 83.4%.

- Optimal augmentations: This achieves a test accuracy of 88.3%.

- Initialization augmentations: This achieves a test accuracy of 80.1%.

**Analysis.** As seen, our method, with about 2-3% of the compute time and significantly lower memory cost than BPTT, obtains very comparable performance in this toy domain, and almost matches the performance with the optimal hyperparameter setting. This indicates effective optimization of the hyperparameters.This performance is achieved even when differentiating through a short FT optimization of 1 step.

**Conclusions.** In this toy domain, our method obtains performance very comparable to BPTT and the optimal hyperparameter setting, and has a fraction of the compute and memory cost of BPTT. This suggests that optimizing the augmentations online is not incurring significant short horizon/truncation bias.

# F  Meta-Parameterized Multitask PT: Further Details

We provide further dataset details, experimental details, and results for the multitask PT experiments. All experiments in this section were run on a single NVIDIA V100 GPU.

## F.1  Further dataset details

The transfer learning benchmark we consider is the biological data benchmark from Hu et al. [28] where the prediction problem at both PT and FT is multitask binary classification: predicting the presence/absence of specific protein functions ($y$) given a Protein-Protein Interaction (PPI) network as input (represented as a graph $x$). The PT and FT datasets both contain 88K graphs.

Hu et al. [28] provide open-source code in their paper to download the raw dataset and then pre-process it. The important steps are extracting subgraphs of the PPI networks centered at particular proteins, and then using the Gene Ontology to identify the set of protein functions associated with each of the proteins.

Importantly, the set of protein functions that we predict at PT time and FT time are different. The PT targets represent coarse-grained biological functions, and the FT targets are fine-grained biological functions, which are harder to obtain experimentally and therefore there is interest in predicting them having pre-trained a model on predicting the targets that are more readily obtained. The PT dataset has labels $y \in \{0,1\}^{5000}$, and the FT dataset has labels $y \in \{0,1\}^{40}$.

Hu et al. [28] discuss the importance of appropriate train/validation/test set splitting for this domain. We follow their suggestion and use the *species split*, where the test set involves predicting biological functions for proteins from new species, not encountered at training/validation time.

We refer the reader to Hu et al. [28] for full details on the pre-processing and construction of subgraphs, the nature of the labels, and the splitting strategy for training, validation, and testing.

## F.2  Further experimental details

### F.2.1  Baselines

We include most important details for baselines in Section 5. Here, for the CoTrain + PCGrad baseline we provide further details, and we also include information about another baseline, CoTrain + Learned Task Weights.

**CoTrain + PCGrad details:** In our implementation, we computed gradient updates using a batch of data from $\mathcal{D}_{PT}$ and $\mathcal{D}_{FT}$ separately, averaging the losses across the set of binary tasks in each dataset (5000 for $\mathcal{D}_{PT}$ and 40 for $\mathcal{D}_{FT}$). PCGrad [72] was then used to compute the final gradient update given these two averaged losses. We also experimented with: (1) computing the overall update using all 5040 tasks (rather than averaging), but this was too memory expensive; and (2) computing the overall update using an average over the 5000 PT tasks and each of the 40 FT tasks individually, but this was unstable and did not converge.

**A further baseline: CoTrain + Learned Task Weights:** We also tried a variant of CoTrain where we learn task weights for each of the 5040 tasks (from $\mathcal{D}_{PT}$ and $\mathcal{D}_{FT}$), along with training the base model. We treat the task weights as high-dimensional supervised learning hyperparameters and optimize these task weights using traditional gradient-based hyperparameter optimization, following the work from [42]. These weights are optimized based on the model's loss on the validation set split of $\mathcal{D}_{FT}$.

### F.2.2  Implementation details

**General details for all methods.**  For all methods, we use the Graph Isomorphism Network (GIN) architecture [69], which was found to be effective on this domain [28].

All methods first undergo PT for 100 epochs with Adam, with a batch size of 32. We used LR=1e-3 for Graph Supervised PT, CoTrain and CoTrain + PCGrad, which is the default LR in the prior work [28]. For the two nested optimization methods that jointly pre-train and learn weights, CoTrain + Learned Task Weights and Meta-Parameterized PT, we used LR=1e-4; we originally tried LR=1e-3, but this led to unstable nested optimization.

After PT, all methods are then fine-tuned for 50 epochs using Adam, with a batch size of 32, over 5 random seeds, using early stopping based on validation set AUC (following [28]). We used 5 seeds rather than 10 (Hu et al. [28] used 10) for computational reasons. For all models, we initialize a new FT network head on top of the PT network body. At FT time, we either FT the whole network (Full transfer) or freeze the PT encoder and learn the FT head alone (Linear Evaluation [50]). We report results here for both FT policies for all methods.

When fine-tuning models using the Full Transfer paradigm, we found that methods were sensitive to LR choices and a FT LR of 1e-3 used in Hu et al. [28] was unstable. The Adam optimizer FT LRs of 1e-5, 3e-5, and 1e-4 were tried for different methods, with FT validation set AUC used to choose the best LR. For Meta-Parameterized PT, we used a full transfer FT LR of 1e-5, and for the other methods, we used 3e-5. For linear evaluation, we used Adam with an LR of 1e-4 for all methods, which was stable.

**Further details for Meta-Parameterized PT.** For meta-parameterized PT, during the meta-PT phase, we use the Adam optimizer with a learning rate of 1e-4 for both PT and FT parameters, and use Adam with a LR of 1 for meta-parameters. These values were set based on the methodology in Appendix D. In Algorithm 2, we use a Neumann series with 1 step in evaluating the inverse Hessian for PT, 1 warmup epoch, $P = 10$ PT steps, $K = 1$ FT steps; we did not search over values for these, and these choices were partly influenced by compute considerations (e.g., large $K$ is more memory expensive).

With these settings, meta-parameterized PT on this task takes about 8-9 GB of GPU memory (about twice the memory cost of normal PT, which is 4-5 GB), and takes about 5 hours to run (as compared to about 2.5 hours for standard PT).

**Further details for CoTrain + Learned Task Weights.** Following a similar process to the above, we used Adam with LR of 1e-4 for the base parameters and LR of 1 for the task weights. We use a Neumann series with 1 step when using the method from [42] for the fairest comparison with meta-parameterized PT.

### F.2.3 Experimental Setup

We re-state the two settings considered, and provide more details about an additional scenario in the Partial FT Access setting.

(1) *Full FT Access:* Provide methods full access to $\mathcal{D}_{PT}$ and $\mathcal{D}_{FT}$ at PT time ($\mathcal{D}_{FT}^{(Meta)} = \mathcal{D}_{FT}$) and evaluate on the full set of 40 FT tasks.

(2) *Partial FT Access:* Consider two situations. First, construct a scenario where we limit the FT data available at PT time directly: $\left|\mathcal{D}_{FT}^{(Meta)}\right| = 0.5\,|\mathcal{D}_{FT}|$. We assess performance on the full FT dataset, as before. Results for this were not presented in the main text due to space constraints.

Second, limit the number of FT tasks seen at PT time, by letting $\mathcal{D}_{FT}^{(Meta)}$ include only 30 of the 40 FT tasks. At FT time, models are fine-tuned on the held-out 10 tasks not in $\mathcal{D}_{FT}^{(Meta)}$. We use a 4-fold approach where we leave out 10 of the 40 FT tasks in turn, and examine performance across these 10 held-out tasks, over the folds.

### F.3 Further results

### F.3.1 Quantitative Results

**Summary of main quantitative results.** Table 4 summarizes the main results across full and limited data/task regimes, reporting the better of Full Transfer and Linear Evaluation. We observe consistent improvements with the meta-parameterized PT strategy over the baselines on the three different experimental evaluation settings.

In the remainder of this section, we discuss these quantitative results further, showing both full transfer and linear evaluation results, and other analysis.

| Method | AUC ($\left\|\mathcal{D}_{\text{FT}}^{(\text{Meta})}\right\| = \|\mathcal{D}_{\text{FT}}\|$) | AUC ($\left\|\mathcal{D}_{\text{FT}}^{(\text{Meta})}\right\| = 0.5\|\mathcal{D}_{\text{FT}}\|$) | AUC ($\mathcal{D}_{\text{FT}}^{(\text{Meta})}$ excludes tasks) |
|---|---|---|---|
| No PT | $66.6 \pm 0.7$ | $66.6 \pm 0.7$ | $65.8 \pm 2.5$ |
| Graph Sup PT | $74.7 \pm 0.1$ | $74.7 \pm 0.1$ | $74.8 \pm 1.8$ |
| CoTrain | $70.2 \pm 0.3$ | $71.0 \pm 0.2$ | $69.3 \pm 1.8$ |
| CoTrain + PCGrad | $69.4 \pm 0.2$ | $71.1 \pm 0.2$ | $68.1 \pm 2.3$ |
| Meta-Parameterized PT | $\mathbf{78.6 \pm 0.1}$ | $\mathbf{78.2 \pm 0.1}$ | $\mathbf{77.0 \pm 1.3}$ |

Table (4) **Meta-Parameterized PT improves predictive performance in three evaluation settings.** Table showing mean AUC and standard error on mean for three different evaluation settings. **First results column: Full FT Access, with evaluation on all tasks, with all FT data provided at PT time. Second results column: Partial FT Access, evaluation with limited FT data at PT time.** When only 50% of the FT dataset is provided at PT time, Meta-Parameterized PT can again improve on other methods in mean test AUC over 40 FT tasks, demonstrating sample efficiency. **Third results column: Partial FT Access, evaluation on new, unseen tasks at FT time.** When 10 of the 40 available FT tasks are held-out at PT, over four folds (each set of 10 FT tasks held out in turn), considering mean test AUC across tasks and folds (and standard error on the mean), meta-parameterized PT obtains the best performance: it is effective even with partial information about the downstream FT tasks.

| Method | Full Transfer | Linear Evaluation |
|---|---|---|
| Rand Init (from [28]) | $64.8 \pm 0.3$ | N/A |
| Rand Init (reimplement, lower FT LR) | $66.6 \pm 0.7$ | N/A |
| Graph Sup PT (from [28]) | $69.0 \pm 0.8$ | N/A |
| Graph Sup PT (reimplement, lower FT LR) | $73.9 \pm 0.2$ | $74.7 \pm 0.1$ |
| CoTrain | $70.2 \pm 0.3$ | $65.9 \pm 0.1$ |
| CoTrain + PCGrad | $69.4 \pm 0.2$ | $62.4 \pm 0.3$ |
| CoTrain + Learned Task Weights | $67.7 \pm 0.2$ | $64.4 \pm 0.1$ |
| Meta-Parameterized PT | $74.7 \pm 0.3$ | $78.6 \pm 0.1$ |

Table (5) **Meta-Parameterized PT results in improved predictive performance.** Table showing mean AUC and standard error on mean across 40 FT tasks on the held-out test set, over 5 random FT seeds. We observe that Meta-Parameterized PT outperforms other baselines in both Full Transfer and Linear Evaluation settings, with significant improvement with Linear Evaluation. Note that with a lower FT LR, baselines from [28] are improved relative to previously reported performance.

**Further results for Full FT Access setting.** Table 5 presents results for all methods across 40 FT tasks, considering both full transfer and linear evaluation. We observe that meta-parameterized PT improves on other baselines in both settings, but most noticeably so in linear evaluation. We also present the results for No PT and Graph Supervised PT from Hu et al. [28]. We observe improvements with our re-implementation, which uses lower FT LRs.

**Studying potential overfitting in CoTrain strategies.** For methods leveraging the FT dataset during PT, the process of performing FT might worsen performance if the model overfits the FT training set. We evaluate FT test performance 'online' during the PT phase, with results in Table 6, and observe that meta-parameterized PT outperforms other methods here also. We do observe some of this overfitting behaviour: note the improved performance on the test set with the learned weights strategy.

**Further results for Partial FT Access setting.** Table 7 shows improved performance even with smaller meta-FT datasets, and Table 8 shows improved performance even with limited tasks at meta-FT time.

### F.3.2 Qualitative Results

We now analyze other aspects of meta-parameterized PT.

**Analyzing learned representations.** To understand the impact of meta-parameterized PT on what the model learns, we compare the learned representations on the FT data across the different PT

| Method | Test AUC | Validation AUC |
|---|---|---|
| Meta-Parameterized PT | 76.1 | 88.2 |
| CoTrain | 67.3 | 83.1 |
| CoTrain + PCGrad | 69.0 | 84.0 |
| CoTrain + Learned Task Weights | 70.7 | 84.6 |

Table (6)   Mean AUC across FT tasks evaluated during PT, for methods that use the FT set at PT time. The separate FT stage may worsen performance of some of the methods, and evaluating in this manner helps account for that. In this setting also, meta-parameterized PT improves on other baselines, in both test and validation set performance.

| Method | Full Transfer | Linear Evaluation |
|---|---|---|
| Rand Init (from [28]) | $64.8 \pm 0.3$ | N/A |
| Rand Init (reimplement, lower FT LR) | $66.6 \pm 0.7$ | N/A |
| Graph Sup PT (from [28]) | $69.0 \pm 0.8$ | N/A |
| Graph Sup PT (reimplement, lower FT LR) | $73.9 \pm 0.2$ | $74.7 \pm 0.1$ |
| CoTrain | $71.0 \pm 0.2$ | $64.4 \pm 0.1$ |
| CoTrain + PCGrad | $71.1 \pm 0.2$ | $64.4 \pm 0.1$ |
| CoTrain + Learned Task Weights | $66.0 \pm 0.3$ | $64.6 \pm 0.3$ |
| Meta-Parameterized PT | $74.3 \pm 0.2$ | $78.2 \pm 0.1$ |

Table (7)   **Meta-Parameterized PT also improves predictive performance with smaller MetaFT datasets.** In a setting where only 50% of the FT dataset is provided at PT time, Meta-Parameterized PT can again improve on other methods in mean test AUC over 40 FT tasks, indicating that it is effective even with limited amounts of FT data available at PT time.

strategies using Centered Kernel Alignment (CKA) [36, 10] in Figure 3. We observe that Meta-Parameterized PT most closely resembles a combination of CoTrain + Learned Weights and Supervised PT, which is sensible given that it blends aspects of both approaches.

**Analyzing learned weights.**    Figure 4 compares learned weights for meta-parameterized PT and the CoTrain+Learned Weights strategies. We observe differences in the histogram of weights, and also the specific values on a per-task basis for these two strategies, indicating that they learn different structures.

**Analyzing negative transfer.**    Figure 5 assesses potential negative transfer on a per-task basis, comparing performance with PT to performance after supervised PT and meta-parameterized PT. Both PT strategies have little negative transfer, and meta-parameterized PT obtains a small extra reduction in negative transfer over standard supervised PT.

| Method | Full Transfer | Linear Evaluation |
|---|---|---|
| Rand Init | $65.8 \pm 2.5$ | N/A |
| Graph Sup PT | $71.5 \pm 1.6$ | $74.8 \pm 1.8$ |
| CoTrain | $69.3 \pm 1.8$ | $67.0 \pm 2.0$ |
| CoTrain + PCGrad | $67.1 \pm 1.5$ | $68.1 \pm 2.3$ |
| CoTrain + Learned Weights | $65.4 \pm 2.0$ | $69.1 \pm 2.6$ |
| Meta-Parameterized PT | $71.3 \pm 2.5$ | $77.0 \pm 1.3$ |

Table (8)  **When evaluating on new, unseen tasks at FT time, meta-parameterized PT again improves on other methods.** We consider a setting where 10 of the 40 available FT tasks are held-out at PT, and only provided at FT time. Over four folds (where different sets of 10 FT tasks are held out in turn), considering mean test AUC across tasks and folds (and standard error on the mean over folds), meta-parameterized PT obtains the best performance. This suggests that the method can perform well even with partial information about the downstream FT tasks.

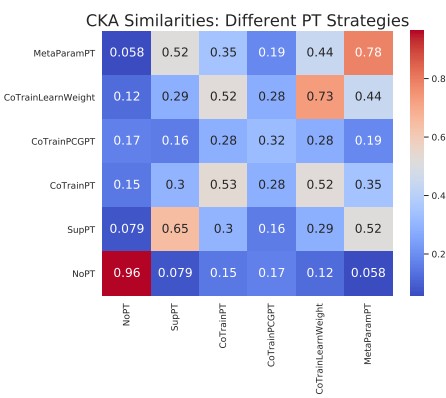

Figure (3)  Comparing learned representations with different PT strategies using CKA [36]. We obtain model representations before the final linear layer across 6400 FT data points, and then compute CKA between pairs of models (averaging over different random initialisations). We observe that Meta-Parameterized PT most closely resembles a combination of CoTrain + Learned Weights and Supervised PT, which is sensible given that it blends aspects of both approaches: meta-parameterized PT learns task weights to modulate the learned representations (as in CoTrain + Learned Weights), and representations are adapted using the PT task alone (as in supervised PT). Interestingly, CoTrain + PCGrad has comparatively little similarity to most other methods in terms of its learned representations.

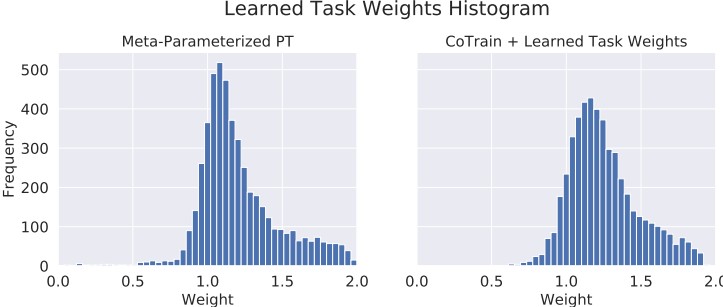

Figure (4)  Comparing learned weights on 5000 PT tasks for meta parameterized PT and with CoTrain + Learned Weights. We observe that different structures appear to be learned by these approaches; the median/half IQR in absolute difference in learned weights is $0.13 \pm 0.09$. Meta-Parameterized PT appears to have more tasks downweighted (weights below 0.5) than the CoTrain approach.

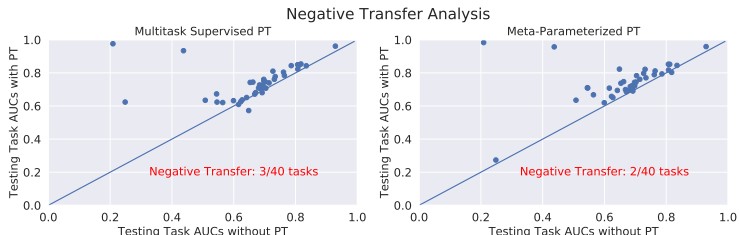

Figure (5)  Comparing performance on FT tasks with and without two different PT strategies: standard supervised PT on the left, and meta-parameterized PT on the right. We show the mean performance over 5 seeds on each of the 40 FT tasks without PT (x axis) and with PT (y axis). A small improvement is observed in reduced negative transfer with Meta-Parameterized PT.

# G   Meta-Parameterized SimCLR PT: Further Details

Figure (6)   A single lead (or channel) of the 12 lead ECG signal and two augmented views (following cropping, jittering, and temporal warping) that are used in contrastive learning.

We provide further SimCLR details, dataset details, experimental details, and results for the SimCLR ECG experiments. All experiments in this section were run on a single NVIDIA V100 GPU.

## G.1   SimCLR summary

SimCLR is a variant of contrastive self-supervised learning [65, 68, 50, 26]. During training, examples are augmented in two different ways to create two views $x_i$ and $x_j$, each of which are encoded independently to produce representations $f^{(\text{enc})}(x_i) = h_i$ and $f^{(\text{enc})}(x_j) = h_j$. These representations are further transformed using a multi-layer decoder ("projection head") to produce vectors $f^{(\text{dec})}(h_i) = z_i$ and $f^{(\text{dec})}(h_j) = z_j$. Models are trained to minimize the normalized temperature-scaled cross-entropy loss (NT-Xent), which contrasts the similarity between pairs of views derived from the same example against the other $2N - 2$ views in a minibatch of size $N$:

$$\mathcal{L}_{\text{PT}}(z_i, z_j) = -\log \frac{\exp(\text{sim}(z_i, z_j)/\tau)}{\sum_{k=1}^{2N} \mathbb{1}_{[k \neq i]} \exp(\text{sim}(z_i, z_k)/\tau)} \tag{5}$$

where $\text{sim}(a, b) = a^\mathsf{T} b / (\|a\|\|b\|)$ is cosine similarity and $\tau$ is the temperature hyperparameter.

## G.2   Further dataset details

We construct our semi-supervised learning (SSL) problem using PTB-XL [64, 20], an open-source dataset of electrocardiogram (ECG) data. Let the model input at both PT and FT time be denoted by $x$, which represents a 12-lead (or channel) ECG sampled at 100 Hz for 10 seconds resulting in a $1000 \times 12$ signal. An example signal is in Figure 6. The PTB-XL dataset contains 21837 ECGs from 18885 unique patients. Each ECG has a 5-dimensional label $y \in \{0, 1\}^5$, where each dimension indicates whether the signal contains certain features indicative of particular diseases/pathologies, namely: Normal ECG, Myocardial Infarction, ST/T Change, Conduction Disturbance, and Hypertrophy. The dataset is split in 10 folds on a patient-level (ECGs from the same patient are all in the same fold), with a suggested train-validation-testing split.

To form an SSL problem from this dataset, we take the training and validation folds, remove the labels, and use only the unlabelled ECGs as the PT dataset. This PT dataset has 19634 unique ECGs. For the FT dataset, we take a random sample of $|\mathcal{D}_{\text{FT}}|$ ECG-label pairs from the training and validation folds. As is common in prior SSL work, we consider different sizes of $\mathcal{D}_{\text{FT}}$ to understand performance given different amounts of labelled data. The FT testing set is the testing fold of the original dataset, which has 2203 ECG-label pairs.

At both PT and FT time, ECGs are normalized before input to the model using zero mean-unit variance normalization, following Wagner et al. [64].

We refer the reader to the open-source data repository on PhysioNet [20], and the paper introducing the dataset [64] for further details.

### G.3 Further experimental details

#### G.3.1 ECG Data Augmentations

To augment each ECG for SimCLR, we apply three transformations in turn (based on prior work in time series augmentation [30, 66]):

1. **Random cropping:** A randomly selected portion of the signal is zeroed out. We randomly mask up to 50% of the input signal.
2. **Random jittering:** IID Gaussian noise is added to the signal. The noise is zero mean and has standard deviation equal to 10% of the standard deviation of the original signal.
3. **Random temporal warping:** The signal is warped with a random, diffeomorphic temporal transformation. To form this, we sample from a Gaussian with zero mean, and a fixed variance at each temporal location, to generate a 1000 dimensional random velocity field. This velocity field is then integrated (following the scaling and squaring numerical integration routine used by Balakrishnan et al. [4, 5]). This resulting displacement field is then smoothed with a Gaussian filter to generate the smoothed temporal displacement field, which is 1000 dimensional. This field represents the number of samples each point in the original signal is translated in time. The field is then used to transform the signal, translating each channel in the same way (i.e., the field is the same across channels).

Two augmented views of an ECG are shown in Figure 6.

#### G.3.2 Implementation details

**General details for all methods.** For all methods, we use a 1D CNN based on a ResNet-18 [23] architecture as the base model that undergoes PT & FT. This model has convolutions with a kernel size of 15, and stride 2 (set based on the rough temporal window we wish to capture in the signal). The convolutional blocks have 32, 64, 128, and 256 channels respectively. The output of these layers is average pooled in the temporal dimension, resulting in a 256 dimensional feature vector. For SimCLR PT, the projection head takes this 256 dimensional vector as input and is a fully connected network with 1 hidden layer of size 256, and output size of 128, with ReLU activation. These hyperparameters were not tuned.

The SimCLR methods are first pre-trained on the PT dataset using SimCLR PT, with a temperature of 0.5 in the NT-Xent loss. We use Adam with an LR of 1e-4 for SimCLR PT, with a batch size of 256, and pre-train for 50 epochs. We consider 3 PT seeds. The methods are then fine-tuned on the FT dataset, replacing the projection head with a new linear FT network head. This whole network is fine-tuned for 200 epochs with Adam, learning rate of 1e-3, batch size of 256. We used an 80%-20% split of the labelled data to form training and validation sets, and validation set AUC was used for early stopping. We consider 5 FT seeds, resulting in a total of 15 runs for each method at each setting.

**Further details for Meta-Parameterized PT.** Meta-parameterized SimCLR incorporates a learned per-example temporal warping strength. We form this by instantiating a four-layer 1D CNN $w(\boldsymbol{x}; \phi)$ that takes in the input ECG $\boldsymbol{x}$ and outputs the variance (1-D output) of the velocity field used to generate the random velocity field. This network has four blocks of convolution, batch norm, and ReLU activation with a kernel size of 15, stride of 2, and 32 channels. We also a optimize a global warping strength scale that multiplies the network output to adjust the overall scale of the warping. The network weights and the global scale are optimized using Adam, with LR=1e-4 and LR=1 respectively. These values were set based on the methodology in Section D. In Algorithm 2, we use a Neumann series with 1 step when evaluating the inverse Hessian for PT, 1 warmup epoch, $P = 10$ PT steps, $K = 1$ FT steps; we did not search over these, and chose these values based on compute considerations. However, we do conduct a comparison with running for other values of $K$ in Appendix G.4.

With these settings, meta-parameterized PT on this task takes about 8-9 GB of GPU memory (about twice the memory cost of normal PT, which is 4-5 GB), and takes about 3 hours to run (as compared to about 1.5 hours for standard PT).

When running meta-parameterized PT with very small meta-FT datasets, of size 10 or 25, the 80%-20% split is not as practical. When $\left| \mathcal{D}_{\text{FT}}^{(\text{Meta})} \right| = 10$, we use a 50-50 split in training and validation, and when it is 25, we use a 60-40 split.

|  | Test AUC at different FT dataset sizes $|\mathcal{D}_{\text{FT}}|$ | | | |
| --- | --- | --- | --- | --- |
|  | 100 | 250 | 500 | 1000 |
| No PT | $71.5 \pm 0.7$ | $76.1 \pm 0.3$ | $78.7 \pm 0.3$ | $82.0 \pm 0.2$ |
| SimCLR | $74.6 \pm 0.4$ | $76.5 \pm 0.3$ | $79.8 \pm 0.3$ | $82.2 \pm 0.3$ |
| SimCLR + OptSLA | $74.6 \pm 0.6$ | $77.0 \pm 0.3$ | $79.6 \pm 0.4$ | $82.8 \pm 0.2$ |
| Meta-Parameterized SimCLR | $\mathbf{76.1 \pm 0.5}$ | $\mathbf{77.8 \pm 0.4}$ | $\mathbf{81.7 \pm 0.2}$ | $\mathbf{84.0 \pm 0.3}$ |

Table (9) **Meta-Parameterized SimCLR obtains improved semi-supervised learning performance.** Table showing mean AUC/standard error over seeds across 5 FT binary classification tasks for baselines and meta-parameterized SimCLR at different sizes of $\mathcal{D}_{\text{FT}}$, with $\mathcal{D}_{\text{FT}}^{(\text{Meta})} = \mathcal{D}_{\text{FT}}$. We observe improvements in performance with meta-parameterized SimCLR over other baselines, including SimCLR + OptSLA, which optimizes the augmentations purely for one-stage supervised learning (rather than two-stage PT and FT).

**An additional baseline: SimCLR + Optimized Supervised Learning Augmentations (SimCLR + OptSLA):** We also investigated a baseline in this domain where the same parametric augmentation policy used for meta-parameterized PT above is: (1) optimized for supervised learning on the labelled FT set, $\mathcal{D}_{\text{FT}}$, following the method from [42]; (2) used as is in SimCLR PT to learn representations; (3) evaluated in a standard FT setting. When using the algorithm from [42], the augmentation meta-parameters are optimized based on the model's loss on the validation set split of $\mathcal{D}_{\text{FT}}$, as is typical in hyperparameter optimization. This baseline compares how optimizing augmentations over the two stage PT and FT compares to optimizing for supervised learning alone. We use Adam for optimization, and use 1 Neumann step in the algorithm from [42].

### G.3.3 Experimental Setup

We re-state the two experimental settings considered. In both settings, we evaluate performance as average AUC across the 5 binary classification tasks, reporting mean and standard error over the 15 runs.

(1) *Full FT Access*, standard SSL: consider different sizes of the labelled FT dataset $\mathcal{D}_{\text{FT}}$ and make all the FT data available at meta-PT time, $\mathcal{D}_{\text{FT}}^{(\text{Meta})} = \mathcal{D}_{\text{FT}}$.

(2) *Partial FT Access*, examining data efficiency of our algorithm: SSL when only limited FT data is available at meta-PT time: $\mathcal{D}_{\text{FT}}^{(\text{Meta})} \subseteq \mathcal{D}_{\text{FT}}$.

### G.4 Further results

We now present additional results in the semi-supervised learning domain.

**Further Full PT Access results with new baseline.** We first present results in the Full PT Access setting, varying $|\mathcal{D}_{\text{FT}}|$ and setting $\mathcal{D}_{\text{FT}} = \mathcal{D}_{\text{FT}}^{(\text{Meta})}$, shown in Table 9. As seen, meta-parameterized SimCLR obtains improvements over the one-stage hyperparameter learning baseline, SimCLR + OptSLA, suggesting that learning augmentations for the two-stage PT & FT process is advantageous.

**Impact of $K$.** We now study the impact of different values of $K$, the number of unrolled differentiation steps when computing the gradient through FT. In our main experiments, we set $K = 1$ for simplicity and computational efficiency. We now seek to understand the following alternative choices:

- $K = 0$: In this setting, we perform **no** FT when optimizing the meta-parameters; that is, we use a randomly initialized linear classifier on top of the PT representations when we compute the FT loss. The meta-learning problem here corresponds to learning PT meta-parameters that optimize the performance of a randomly initialized linear classifier. This experiment tests what happens when the gradient through FT is noisy, but the component through PT is informative.

- $K > 1$: This setting tests whether unrolling more steps during FT can improve the gradient signal received when optimizing meta-parameters.

| | Test AUC at different FT dataset sizes $|\mathcal{D}_{\text{FT}}|$ | | | |
|---|---|---|---|---|
| | 100 | 250 | 500 | 1000 |
| SimCLR | $74.6 \pm 0.4$ | $76.5 \pm 0.3$ | $79.8 \pm 0.3$ | $82.2 \pm 0.3$ |
| $K = 0$ | $75.3 \pm 0.5$ | $77.1 \pm 0.5$ | $80.5 \pm 0.4$ | $83.7 \pm 0.3$ |
| $K = 1$ | $76.1 \pm 0.5$ | $77.8 \pm 0.4$ | $81.7 \pm 0.2$ | $84.0 \pm 0.3$ |
| $K = 5$ | $76.6 \pm 0.2$ | $78.3 \pm 0.3$ | $81.9 \pm 0.4$ | $84.2 \pm 0.2$ |
| $K = 10$ | $76.3 \pm 0.5$ | $78.1 \pm 0.4$ | $81.7 \pm 0.4$ | $84.3 \pm 0.3$ |

Table (10)  **Examining how the number of unrolled FT steps affects semi-supervised learning perfor-mance.** Table showing mean AUC/standard error over seeds across 5 FT binary classification tasks for meta-parameterized SimCLR when we vary the number of unrolled FT steps used to compute the meta-parameter gradient. We observe that using a noisy FT gradient ($K = 0$) improves on not optimizing augmentations at all, but is worse than using a single step ($K = 1$). Using more unrolled steps can lead to small improvements.

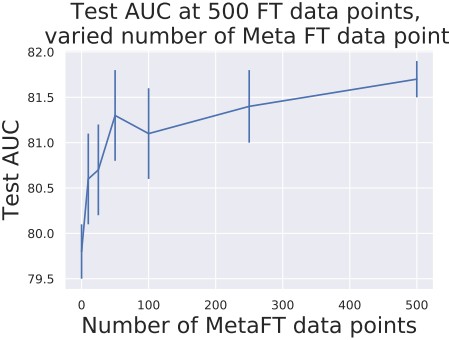

Figure (7)  **Meta-Parameterized SimCLR is effective when only small amounts of FT data are available at PT time.** Test set AUC when varying $\left|\mathcal{D}_{\text{FT}}^{(\text{Meta})}\right|$: the number of FT data points provided at PT time, considering $|\mathcal{D}_{\text{FT}}| = 500$. We see that meta-parameter learning is effective even at small $\left|\mathcal{D}_{\text{FT}}^{(\text{Meta})}\right|$ (sharp improvement in performance at small $\left|\mathcal{D}_{\text{FT}}^{(\text{Meta})}\right|$)

Results are shown in Table 10. As can be seen, unrolling through one step of FT ($K = 1$) improves upon using a noisy FT gradient ($K = 0$) in all cases. Using a noisy FT gradient but an informative PT component ($K = 0$) improves on not optimizing the augmentations at all (SimCLR). This implies that both the PT and FT components inform the optimization of the augmentations. When $K > 1$, we do observe some improvement with more steps, but diminishing returns as $K$ increases further.

**Further Partial PT Access results.**   We now consider the Partial FT Access setting. Firstly, Figure 7 shows the performance of meta-PT when we fix $|\mathcal{D}_{\text{FT}}| = 500$ and vary $\left|\mathcal{D}_{\text{FT}}^{(\text{Meta})}\right|$. We find that meta-PT can be effective even with very small validation sets (consider the sharp improvement at small MetaFT data points, with 0 MetaFT points representing no optimization of the augmentations). This result was just considering the $|\mathcal{D}_{\text{FT}}| = 500$ setting; in Figure 8, we consider other FT dataset sizes and analyze performance. We see that in all regimes, there is a noticeable increase in performance at small meta-FT dataset sizes, which is a desirable result since it shows that our algorithm can be effective even with very limited labelled data available at meta-PT time.

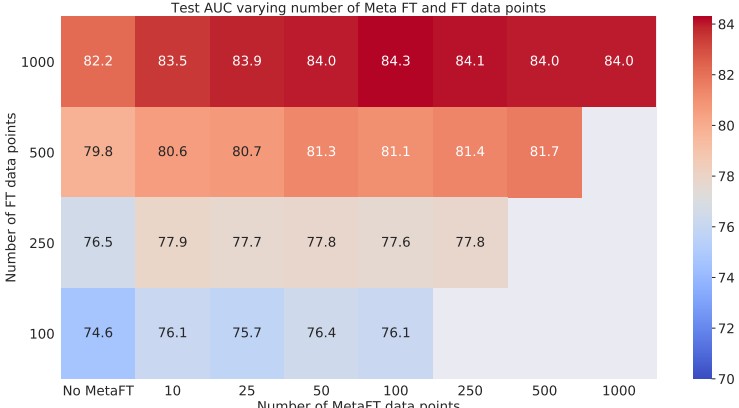

Figure (8)     Sweeping over meta FT/FT data points and analyzing performance trends. We observe that across various settings of FT data availability, a small amount of MetaFT data can lead to significant performance improvements.