# OpenReview forum: "Meta-learning to Improve Pre-training"
_NeurIPS.cc/2021/Conference — NeurIPS 2021 Poster_

### Official Review · Reviewer_1DnG · 2021-07-15

**Rating:** 6
**Confidence:** 3

**Summary:**

This paper introduces a gradient-based algorithm which improve the first stage of two-stage pre-trained and fine-tune process with meta-learning techniques. The major contribution is proposing to use additional parameters called meta-parameters to augment pre-trained stage and use a novel gradient-based algorithm to learn meta-parameters through the two-stage process. The experimental results demonstrate the significant improvement of the proposed approach over the standard two-stage PT and FT process.

**Ethical Concerns:**

No ethical concerns.

**Limitations And Societal Impact:**

No societal impact.

**Main Review:**

== Pros ==
- Using meta-parameters to improve pre-trained model has enough novelty, and further research on this open problem is meaningful.
- The authors propose a novel gradient-based algorithm to optimize meta-parameters.
- The proposed approach achieves significant improvements on both tasks.

== Cons ==
- Meta-parameters as neural networks could have much more representative power than weight of task or temporal transformation, I'd like to see more complex scenarios where this approach can be applied.
- Despite the experiment setting in this paper may be new, three still need some other meta-learning techniques as baseline if possible. Because as far as I know, there are many gradient-based meta-learning algorithms that are not used in PT and FT process for now.
- Meta-learning techniques usually requires much more computing resources and time, an analysis on GPU memory and execution speed would be appreciated.
- Some symbols may be confused for me, such as line 4 in Algorithm 1.

**Time Spent Reviewing:**

8

---

> ### Author Response · Authors · 2021-08-10
> **Author's response to review**
>
> We thank you for your comments and time!
>
> Below, we provide detailed responses to each of the 4 cons identified in your review. In summary, we
>
> 1. Provide clarification on the use of meta-parameters as neural networks, highlighting the two experimental settings in our paper that consider this case.
>
> 2. Provide more details on meta-learning baselines including results for two meta-learning baselines which we compare against (one of which is currently in the appendix of the paper and the other is new).
>
> 3. Discuss the computational considerations of our algorithm, including further comments on time and memory cost.
>
> 4. Note that we will update the notation in the specified line to make it clearer.
>
> We look forward to hearing your response to these key points. If you feel we have not sufficiently addressed your concerns to motivate increasing your score, we would love to hear from you further on what points of concern or confusion remain and how we can improve the work in your eyes. Thank you again!
>
> # Detailed response:
>
> ## Con 1: On meta-parameters as neural networks
>
> #### **Re:** *"Meta-parameters as neural networks could have much more representative power than weight of task or temporal transformation, I'd like to see more complex scenarios where this approach can be applied."*
>
> In the experiments in our paper, we consider two settings where this is the case.
>
>   1. In Section 6 (meta-parameterized SimCLR experiments) we define meta-parameters to be the (high-dimensional) weights of a neural network that is trained to output a per-example augmentation strength.
>
>   2. In our second synthetic domain where we learn example importance weighting for MNIST, we instantiate meta-parameters as the weights of the neural network that outputs the per-example importance weight.
>
>
> ## Con 2: On other meta-learning baselines
>
> **Re:** *"Despite the experiment setting in this paper may be new, three still need some other meta-learning techniques as baseline if possible. Because as far as I know, there are many gradient-based meta-learning algorithms that are not used in PT and FT process for now."*
>
> First, we clarify that many other meta-learning baselines do not apply in our setting; in fact, most existing meta-learning algorithms cannot directly be used in the PT and FT setting because they are not constructed to differentiate through this two-stage optimization process. This necessitates the development of our new algorithm.
>
> Additionally, we do compare with a meta-learning method in our GNN experiments, namely CoTrain + Learned Weights. Please see Appendix E.2 for further details on the method, and quantitative results for this method in Tables 5-7, Appendix E. We find that our approach significantly outperforms this hyperparameter learning baseline, and also study a potential reason for this in Table 6.
>
> Since the submission, we have also evaluated another meta-learning baseline on the meta-parameterized SimCLR domain. We studied a situation where we:
>
> * Learn the augmentation parameters for supervised learning on the finetuning set $\mathcal{D}_{\text{FT}}$, following the method from [1].
>
> * Perform SimCLR PT with these learned augmentation parameters
>
> * Perform fine-tuning, and then evaluate on the FT test set.
>
> Note that this approach could be considered an ablation where we learn augmentations for a traditional supervised learning problem and see how these compare to augmentations directly optimized for semi-supervised learning, over a two-stage learning problem.
>
> The results are as follows. Each column of results is the test AUC at different FT dataset sizes.
>
>
>
> |                                  	|    100    	|    250    	|     500    	|    1000   	|
> |----------------------------------	|:---------:	|:---------:	|:----------:	|:---------:	|
> | Meta-Parameterized SimCLR        	| 76.1+-0.5 	| 77.8+-0.4 	| 81.7+- 0.2 	| 84.0+-0.3 	|
> | Hyperparameter learning baseline 	| 74.6+-0.6 	| 77.0+-0.3 	| 79.6+-0.4  	| 82.8+-0.2 	|
>
>
>
> As can be seen, learning hyperparameters over the PT and FT stages together improves performance.
>
>
> ## Con 3: On computational considerations
>
> #### **Re:** *"Meta-learning techniques usually requires much more computing resources and time, an analysis on GPU memory and execution speed would be appreciated."*
>
> Thank you for raising this important point. As discussed in Section 3, in the paragraph: ‘Meta-PT in practice’,  with the specified settings and only unrolling the FT head during hyperparameter learning, our algorithm has about twice the time cost as normal pre-training. In the GNN domain, this means that meta-parameterized PT takes about 5 hours (compared to about 2.5 hours for standard PT). In the semi-supervised learning domain, this means that meta-parameterized PT takes about 3 hours (compared to about 1.5 hours for standard PT). This represents a significant time improvement over traditional hyperparameter optimization, which may require performing PT and FT many tens or hundreds of times.
>
> In terms of memory cost, our simple implementation has about twice the memory cost as running normal PT (8-9 GB of GPU memory in both settings, compared to about 4-5 GB during normal PT). When unrolling only the head of the network during FT, this implementation can be improved to obtain about 1.5x the memory cost, if vector-Jacobian products in the algorithm (lines 9 and 10 in Algorithm 1) are computed without instantiating the Jacobian matrices explicitly.
>
>
> ## Con 4: On notational clarity
>
> #### **Re:** *"Some symbols may be confused for me, such as line 4 in Algorithm 1."*
>
> Thank you for pointing this out, and we apologize for any confusion here. We will fix this line to more clearly identify the parameter vector that is being updated.

---

> ### Author Response · Authors · 2021-08-23
> **Author response to reviewer**
>
> Dear Reviewer,
>
> Thank you once again for your review. In our response, we believe we have replied to your main points regarding: the use of meta-parameters as neural networks, meta-learning baselines, computational cost of our method, and notational clarity. We would greatly appreciate it if you could update your review, given our response.
>
> Thank you again for your time!

---

### Official Review · Reviewer_q25x · 2021-07-15

**Rating:** 5
**Confidence:** 2

**Summary:**

The paper describes a gradient-based hyperparameter optimization for differentiable pre-training hyperparameters in a pre-training and fine-tuning setup. The proposed approach approximately differentiates through the two-stage learning process. The authors show the benefit of their approach in two experiments, where they outperform multiple baselines.

**Limitations And Societal Impact:**

The authors have adequately addressed the limitations and potential negative social impact of their work.

**Main Review:**

Originality: In my opinion, this kind of meta-learning on two-stage setups is not new (see ref. [67] and [68]) but the more general approach is novel. It feels a bit like MAML [15] for differentiable hyperparameters (MAML is mentioned in the related work). The difference to related work is described and related work is cited.

Clarity: I had issues following this paper. While in the abstract and introduction the authors talk about “hyperparameters” they mention only in the limitations section that the hyperparameters have to be differentiable. When I read hyperparameters of neural network training, I think mainly on parameters like the learning rate. The abstract promises an HPO method for hyperparameters in general without mentioning this limitation.
In the second section in the paragraph “Notation”, the authors introduce uppercase theta, uppercase psi, and uppercase phi as “spaces of parameters for models” without any further description on the difference between these “spaces”.  Two pages later in line 123, the authors name phi “encoder parameters” and psi “decoder parameters” but without mentioning that they assume an encoder-decoder setup.
In the abstract and introduction, the authors write about optimizing neural networks while in the problem setup they describe the more general case of “parametric model”.
I wasn’t able to follow their algorithm due to the lack of knowledge about theta, psi, and phi.
In the experiment section, it is unclear where the definition of the loss comes from and how P and K are tuned.
I would propose to clarify what kind of hyperparameters are addressed by this approach and rework the problem formulation and notation.

Quality: I can’t assess the methods since I didn’t understand the algorithm in detail due to the lack of knowledge about theta, psi, and phi. The method is evaluated in two experiments with good results.

Significance: In my opinion, the experiments are very specialized for a general tone in the abstract and introduction. I couldn't follow the notation easily. Even if I am not a domain expert for gradient-based hyperparameter optimization this shouldn’t be the case in my opinion. In the current state of this paper, I find it hard to make use of the approach.

**Time Spent Reviewing:**

6

---

> ### Author Response · Authors · 2021-08-10
> **Author's response to review**
>
> We thank you for your comments and time!
>
> Below, we provide detailed responses to each of the main points identified in your review. In summary, we
>
> 1. Clarify the originality and novelty of our method.
>
> 2. Address the various points relating to clarity, including scope, notation, and terminology.
>
> 3. Outline why we believe our method to be of broader significance.
>
>
> We look forward to hearing your response to these key points. If you feel we have not sufficiently addressed your concerns to motivate increasing your score, we would love to hear from you further on what points of concern or confusion remain and how we can improve the work in your eyes. Thank you again!
>
>
> # Detailed response
>
> #### **Re:** *"In my opinion, this kind of meta-learning on two-stage setups is not new (see ref. [67] and [68]) but the more general approach is novel. It feels a bit like MAML [15] for differentiable hyperparameters…"*
>
> We highlight that our work is quite different from the traditional literature in nested optimization (including approaches such as MAML), since we must obtain gradients through two stages of optimization (both PT and FT).
>
> Compared to the two-stage setups cited in the work (references [67] and [68] in the paper), our algorithm represents a more general and scalable approach to estimate gradients through PT and FT, without using extensive first-order approximations or relying only on differentiating through the learning process (which may not scale as well).
>
>
> #### **Re:** *"... in the abstract and introduction the authors talk about “hyperparameters” they mention only in the limitations section that the hyperparameters have to be differentiable. When I read hyperparameters of neural network training, I think mainly on parameters like the learning rate. The abstract promises an HPO method for hyperparameters in general without mentioning this limitation."*
>
> Although our method cannot be used to optimize all hyperparameters (such as learning rates), we do clearly mention in our abstract and introduction the nature of hyperparameters we consider, such as task reweighting strategies and augmentation policies.  Furthermore, we feel that the ability to handle important hyperparameters such as task weighting schemes, data augmentation networks, and more (as evidenced by our real-world experiments) is still of significant value.  Given we stated the type of hyperparameters considered early in the paper, we believed the most natural place to comment on the broader scope was the scope/limitations section.
>
> #### **Re:** *"in the paragraph “Notation”, the authors introduce uppercase theta, uppercase psi, and uppercase phi as “spaces of parameters for models” without any further description on the difference between these “spaces”."*
>
> We apologize for any confusion on this point, and will work to ensure our notation is as clear as possible for all reviewers. Further details on the definitions of these spaces are also, we expect, not essential for understanding the method -- they will be isomorphic to some real-valued vector space $\mathbb{R}^n$ in all cases.
>
> #### **Re:** *"the authors name phi “encoder parameters” and psi “decoder parameters” but without mentioning that they assume an encoder-decoder setup."*
>
> We thank the reviewer for this point and apologize for any confusion. We will make it consistent with the earlier terminology of ‘feature extractor’  (encoder)and  ‘head’ (decoder).
>
> #### **Re:** *"the authors write about optimizing neural networks while in the problem setup they describe the more general case of “parametric model”."*
>
> We refer to neural networks in the abstract and introduction since this is the specific class of parametric model our experiments study. In presenting our framework and algorithm, we use more general terminology since our method could apply to other (differentiable) parametric models also.
>
> #### **Re:** *" I wasn’t able to follow their algorithm due to the lack of knowledge about theta, psi, and phi."*
>
> We apologize for any lack of clarity here; we define relevant notation in Section 2, and provide a full glossary of notation in Table 3. We also provide further clarifying details about our method in Appendix B, which are referred to from the main text (lines 126-127). If these were not clear, we can supplement them with additional descriptions for clarity.
>
> #### **Re:** *"In the experiment section, it is unclear where the definition of the loss comes from and how P and K are tuned."*
>
> The loss for the weighted multitask PT setup is defined analogously to the example in Section 2.2 and represents a weighted average cross-entropy loss over tasks. As described in Appendix E and F,  the values of P and K were set based on compute considerations; in particular, large K would lead to higher memory overhead.
>
>
> #### **Re:** *"... the experiments are very specialized for a general tone in the abstract and introduction. … In the current state of this paper, I find it hard to make use of the approach."*
>
> As we describe in our paper, the method can be used to optimize general differentiable pre-training hyperparameters; our various experimental settings demonstrate different applications of this idea: namely, simple data augmentation schemes for supervised learning (Section 4), example level weighting schemes (Section 4), weighted multitask learning (Section 5), and more sophisticated data augmentation schemes for self-supervised learning (Section 6). We therefore respectfully disagree with the comment that the experiments are very specialized.
>
> We note that there are a large range of other hyperparameters that could be optimized with our approach, including: learned auxiliary labels [3], learned attention masks [2], and high-dimensional regularization parameters [1].
>
> ## References:
>
> [1] Optimizing Millions of Hyperparameters by Implicit Differentiation, AISTATS 2020
>
>
> [2] Teaching with Commentaries, ICLR 2021
>
> [3] Auxiliary Learning by Implicit Differentiation, ICLR 2021

---

> > ### Comment · Reviewer_q25x · 2021-08-27
> > **Response to comment**
> >
> > Dear Author's,
> > thanks for your comment and clarification, they helped me to understand the idea and paper better, however, I am unsure if I understood it correctly and if the final paper will be easier to follow. Due to my better understanding now, I increase my score.

---

> > > ### Author Response · Authors · 2021-08-27
> > > **Author reply**
> > >
> > > Dear Reviewer, thank you very much for reading our response and updating your score. We are happy to provide more information about the revised presentation and any points that are still unclear if that would help clarify aspects of the work, and help improve the paper in your eyes?
> > >
> > > Thank you again!

---

> ### Author Response · Authors · 2021-08-23
> **Author note to reviewer**
>
> Dear Reviewer,
>
> Thank you once again for your review. In our response, we believe we have replied to your main points regarding: the novelty of our method, clarity of exposition, and broader significance of our method. We would greatly appreciate it if you could update your review, given our response.
>
> Thank you again for your time!

---

### Official Review · Reviewer_BiAi · 2021-07-15

**Rating:** 6
**Confidence:** 3

**Summary:**

This work seeks to extend gradient-based HPO to the two stage setting of pretraining + finetuning. While the method used isn't original it is used in an original problem set up.

**Main Review:**

-----------------------------------PROS
- The PT+FT paradigm considered is original and relevant
- the paper is mostly clear
- Considering Full FT access and Partial FT access is a good thing

-----------------------------------CONS

MAJOR

- You explain that you update psi_FT online, which may remove memory issues with BPTT, but has been shown to lead to biased (greedy) solutions ("Short horizon bias", Wu2018). These don't converge to hypergradients close to the actual (full horizon) hypergradients we care about. This is a major limitation to your model since the gradients that come from the finetuning stage are in my experience almost as good as random noise, and yet this isn't listed in your Limitations section. A toy model where you can use full-horizon BPTT for both PT and FT would allow you to measure how accurate your algorithm is at approximating hypergradients as the number of steps increase in the PT and FT stage.
- While experiments seem sensible, I feel like the experiments I would need to see to make sure your method is competitive with sota methods aren't included. For instance, I would have liked to see comparisons with sota methods in multi-task learning and/or semi-supervised learning and/or domain adaptation for common image datasets like CIFAR-10 where lots of other methods have been applied. This is because the main difficulties of gradient-based HPO (e.g. gradient degradation) come in for a large number of steps in the PT/FT stages. In line 196 you seem to be using a very short horizon of P=10, which is nowhere near the 10^4 gradient steps you'd need for a CIFAR10-like dataset.
- There is also an issue of novelty, since the combination of implicit differentiation and BPTT is fairly straightforward (just multiplication as per chain rule) and one may argue doesn't really make up a "new" algorithm in itself.

MINOR

- Your toy experiments in section 4 are both experiments that don't require your PT + FT 2-stage method. Indeed, these have been done where the properties you apply to the FT data are simply applied to the validation data instead. It would be nice to have experiments where ONLY your 2-stage approach is sensible.
- I found Figure 1 somewhat confusing. For instance I'm not sure what "Finetuning Data" + "Sample" labels achieve, why the "meta parameters" are connected to "Pre-training" on top of the "Model" and "Pre-train Dataset" labels since potentially the "meta parameters" could include "Model" and "Pre-training dataset". I understood your set up from the equations in 2.1, although a bit more details would have saved me time. For example in line 78 you could explicitely state the the final weights of the PT stage are used as the init weights of the FT stage, such that the final weights of the FT stage minimize the FT loss.

**Time Spent Reviewing:**

7

---

> ### Author Response · Authors · 2021-08-10
> **Author's response to review (part 1)**
>
> We thank you for your comments and time!
>
>
> Below, we provide detailed responses to each of the 5 cons identified in your review. In summary, we
>
> 1. Provide further comments on the potential short-horizon bias problem including citations from related work and novel results using longer optimization trajectories during FT to assess the impact this has on our results.
>
> 2. Clarify that our method works online, and that the two real-world datasets we consider require substantial training times. We also describe the range of baselines we compare against, and provide results for a new meta-learning baseline.
>
> 3. Clarify the technical significance and novelty of our method.
>
> 4. Address the comment on the setup of the toy experiments.
>
> 5. Note that we will address the points on clarity and try to revise Figure 1 to make it easier to understand.
>
>
> We look forward to hearing your response to these key points. If you feel we have not sufficiently addressed your concerns to motivate increasing your score, we would love to hear from you further on what points of concern or confusion remain and how we can improve the work in your eyes. Thank you again!
>
>
>
> # Detailed response
>
>
> ## Con 1: On the short horizon bias problem
>
> #### **Re:** *“...update psi_FT online, which may remove memory issues with BPTT, but has been shown to lead to biased (greedy) solutions ("Short horizon bias", Wu2018). These don't converge to hypergradients close to the actual (full horizon) hypergradients we care about. This is a major limitation to your model”*
>
> Thank you for raising this important concern -- it warranted more commentary in the main text and we will work to improve our clarity on this point.
>
> **In summary:**  We understand short-horizon bias to be a special case of the bias induced by truncating telescoping sums for optimization parameters.  The effects of the truncation can be pronounced with optimization parameters, but there exist methods like [8] to deal with these if they occur.  Prior works of [1-5] did not observe significant truncation bias in similar settings to ours, so we also did not expect this bias to be a significant issue.  We have added additional experiments below to investigate if we have a truncation bias, which show the effect is limited.
>
>
> **A closer look:** There are two hypergradients in our system that could suffer from bias: the PT hypergradients and the FT hypergradients. *In both cases, we feel that any impact from biased hypergradients is minimal, and that the significant performance improvements observed still demonstrate the value of our algorithm.*
>
> We will argue for this claim through each hypergradient term separately.
>
>
> **PT Hypergradient:** The PT hypergradient does not suffer from the short-horizon bias because the PT model is expected to have approximately converged at each hyperparameter update. This is not only a requirement of the implicit function theorem and the algorithm from [1] to apply, but also is directly enforced in our system through the use of online-updates and a warmup period (see full algorithm description in Appendix B of the paper: Algorithm 2, and lines 598-601). As seen in prior work using this method [1-3], computing hypergradients in such an online fashion results in effective learning, so we do not believe this to be a significant issue.
>
>
> **FT Hypergradient:** For the gradient through FT, we acknowledge that differentiating through only one step could, in theory, produce biased hypergradients.
>
> However, prior work on meta-learning structures similar to ours has computed meta-parameter updates in a similar fashion by differentiating through one step of training, and has obtained impressive, near SOTA results [4], [5]. Like these works, we also obtain impressive improvements even with $K=1$.
>
> If short-horizon bias is an issue, then we could use the method from [8] to fix it.  We did not expect there to be a significant bias here, due to [4], [5], but in the interest of ensuring this bias is not a significant concern, we have added experiments showing that the horizon does not significantly affect the result.
>
> In particular, we considered the meta-parameterized SimCLR and ECG data experiments (Section 6) and studied using a larger number of FT optimization steps $K$, namely $K=5$ and $K=10$. The results are as follows, including the results in the paper for $K=0$ (no meta-learning) and $K=1$ for completeness, from Table 2. Each column of results is the test AUC at different FT dataset sizes.
>
>
> |                        	|    100    	|     250    	|    500    	|    1000   	|
> |------------------------	|:---------:	|:----------:	|:---------:	|:---------:	|
> | K=0 (No meta-learning) 	| 74.6+-0.4 	| 76.5+-0.3  	| 79.8+-0.3 	| 82.2+-0.3 	|
> | K=1                    	| 76.1+-0.5 	| 77.8+-0.4  	| 81.7+-0.2 	| 84.0+-0.3 	|
> | K=5                    	| 76.6+-0.2 	| 78.3+-0.3  	| 81.9+-0.4 	| 84.2+-0.2 	|
> | K=10                   	| 76.3+-0.5 	| 78.1 +-0.4 	| 81.7+-0.4 	| 84.3+-0.2 	|
>
>
>
>
> As seen from the results, we do observe some improvement with more steps, but diminishing returns as $K$ increases further.
>
> To summarize, we see that:
> 1. Most of the improvement over the no meta-learning baseline appears to come from just setting $K=1$;
>
> 2.  Running with more $K$ is certainly possible with our algorithm given we only need to unroll the network head;
>
> 3.  Prior work has shown the value of single-step differentiation to compute meta-parameter gradients;
>
> Given all this, we do not see this as a major limitation of our method.
>
>
>
>
> ## Con 2: On optimization horizons, further experiments, and more baselines
>
> #### **Re:** *"... the main difficulties of gradient-based HPO (e.g. gradient degradation) come in for a large number of steps in the PT/FT stages. In line 196 you seem to be using a very short horizon of P=10, which is nowhere near the 10^4 gradient steps you'd need for a CIFAR10-like dataset."*
>
> We apologize for any confusion on this point. Our method actually does not reduce the number of steps for PT & FT in general. As stated in Algorithm 1, we solve the hyperparameter optimization problem in an online fashion, similarly to related work [1,2,3]. In this way, while we do use a small number of PT & FT steps in each inner loop iteration, the total number of steps used during PT and FT is therefore large for both domains, on the order of $10^4$ to $10^5$ steps (lines 212-214, lines 287-292), similar to the CIFAR10-like datasets you suggest. This online approach is significantly more computationally efficient than performing offline updates.
>
>
>
> #### **Re:** *"...I feel like the experiments I would need to see to make sure your method is competitive with sota methods aren't included. … comparisons with sota methods in multi-task learning and/or semi-supervised learning and/or domain adaptation for common image datasets like CIFAR-10....”*
>
> Note that both benchmarks we evaluate on are well-studied and have had significant follow-on work.
>
> In particular, the paper proposing the GNN Biological dataset benchmark [6] itself has an extensive set of baselines, and our best-performing approach outperforms the SOTA in that paper.
>
> In our experiments on GNN PT, we compare to a range of effective baselines, including a variety of multitask learning approaches and another meta-learning-based approach (See Section 5 and Appendix E for a full list of baselines considered).  Specifically, in terms of multitask learning baselines, we evaluate not only standard multitask learning (CoTrain), but also a recently proposed, more effective multi-task learning approach, PCGrad [7], and find that we outperform both significantly (see Tables 1, 4,5,6,7,8).
>
> In the self-supervised learning domain, we consider the main relevant baselines in our paper. Since the deadline, we have also evaluated another baseline where we:
>
>
> * Learn the augmentation parameters for supervised learning on the finetuning set $\mathcal{D}_{\text{FT}}$, following the method from [1].
>
> * Perform SimCLR PT with these learned augmentation parameters
>
> * Perform fine-tuning, and then evaluate on the FT test set.
>
> Note that this approach could be considered an ablation where we learn augmentations for a traditional supervised learning problem and see how these compare to augmentations directly optimized for semi-supervised learning, over a two-stage learning problem.
>
> The results are as follows. Again, each column of results is the test AUC at different FT dataset sizes.
>
>
>
> |                                  	|    100    	|    250    	|     500    	|    1000   	|
> |----------------------------------	|---------	|---------	|----------	|---------	|
> | Meta-Parameterized SimCLR        	| 76.1+-0.5 	| 77.8+-0.4 	| 81.7+-0.2 	| 84.0+-0.3 	|
> | Hyperparameter learning baseline 	| 74.6+-0.6 	| 77.0+-0.3 	| 79.6+-0.4  	| 82.8+-0.2 	|
> As can be seen, learning hyperparameters over the PT and FT stages together improves performance.
>
> Overall, given the fairly extensive nature of our experiments, we respectfully disagree that we need more datasets and comparisons to demonstrate the efficacy of our approach.

---

> > ### Author Response · Authors · 2021-08-10
> > **Author's response to review (part 2)**
> >
> > ## Con 3: On the Novelty of our Method
> >
> > #### **Re:** *"There is also an issue of novelty, since the combination of implicit differentiation and BPTT is fairly straightforward (just multiplication as per chain rule) and one may argue doesn't really make up a "new" algorithm in itself."*
> >
> > We respectfully disagree with the point that our method is too straightforward to be a technical contribution. We believe that the simplicity of the method is a strength -- it is an intuitive way of combining two existing ideas, is well-justified and theoretically motivated, and achieves strong performance on two important real-world domains. Furthermore, to the best of our knowledge, despite its intuitive nature, there is no prior work studying this particular approach, so we do view this as an important contribution.
> >
> > In addition, our approach is quite flexible in that it could theoretically apply to any differentiable hyperparameters/models. There are a large range of other hyperparameters that could be optimized with our approach, including: auxiliary labels [3], learned attention masks [2], and high-dimensional regularization parameters [1]. Given all this, we do not feel this limitation is sufficient to motivate rejection.
> >
> >
> > ## Con 4: On toy experiments
> >
> > #### **Re:** *"Your toy experiments in section 4 are both experiments that don't require your PT + FT 2-stage method. … It would be nice to have experiments where ONLY your 2-stage approach is sensible."*
> >
> > Thank you for this comment. We have additional experiments where we consider a domain-shifted FT dataset (for example, different MNIST digits at PT and FT) and we can mention these in a revision.
> >
> > ## Con 5: On clarity
> > Thank you for the comments on Figure 1 and areas where the presentation lacks clarity. We will try to revise these to make it clearer for readers.
> >
> >
> > ## References:
> >
> > [1] Optimizing Millions of Hyperparameters by Implicit Differentiation, AISTATS 2020
> >
> > [2] Teaching with Commentaries, ICLR 2021
> >
> > [3] Auxiliary Learning by Implicit Differentiation, ICLR 2021
> >
> > [4] Meta Pseudo Labels, CVPR 2021
> >
> > [5] Learning Data Manipulation for Augmentation and Weighting, NeurIPS 2020
> >
> > [6] Strategies for pre-training graph neural networks, ICLR 2020
> >
> > [7] Gradient Surgery for Multi-Task Learning, NeurIPS 2020
> >
> > [8] Efficient Optimization of Loops and Limits with Randomized Telescoping Sums, ICML 2019

---

> > > ### Comment · Reviewer_BiAi · 2021-08-12
> > > **Reply**
> > >
> > > Thank you for your clarification.
> > >
> > > About insufficient experiments, thank you for making your case stronger by pointing to the value of your baselines already in the paper. What I had in mind was an experiment that specifically targeted say a small problem (where both PT+FT fit in memory so you can use BPTT to train them) so as to compare your method to the gold standard (BPTT, i.e. when it fits in memory..). This is because the "Short horizon bias" paper makes a strong claim that online updates do not lead to sensible hypergradients (i.e. people are wrong for doing online HPO) but you make little effort to refute this claim in your setting. Apart from this experiment, do you have any intuition as why, say, meta-learning the learning rate online isn't sensible, but doing FT online in your method is? How is your setting more amenable to online updates? If your PT component is very good and the FT component is poor, could you still observe the performance you're observing? Or do you need both components to be good?
> > >
> > > Re: K>1. In my experience, if your finetuning stage is ~10^3 gradient steps, K=1,5,10 would basically be similarly "local" in terms of meta-optimization, since they are much smaller than the optimal value (i.e. K=10^3). So I don't think these experiments prove or disprove anything.

---

> > > > ### Author Response · Authors · 2021-08-17
> > > > **Response to comment**
> > > >
> > > > Thank you for your reply.  We apologize for not fully understanding your concerns before -- thank you for clarifying these! We respond by first summarizing your concerns as we interpret them, and then replying in more detail to these points and other things you have raised in your comment.
> > > >
> > > > In our main response, we do the following:
> > > >
> > > > * Point to existing experiments in the paper indicating the short horizon bias issue may not be a problem for our method.
> > > >
> > > > * Consider two new experiments to verify this issue is not affecting performance, including the BPTT experiment with synthetic data you suggest, and a setting building on our semi-supervised learning experiments where we find a closed form solution for the FT head.
> > > >
> > > > * Provide an argument as to why optimizing our hyperparameters by differentiating through a small number of optimization steps may not suffer from the issues faced when optimizing the LR in a similar fashion
> > > >
> > > > * Provide new experiments to investigate the relative importance of PT and FT components to our hypergradient.
> > > >
> > > > We look forward to hearing your response to these points. Thank you again!

---

> > > > > ### Author Response · Authors · 2021-08-17
> > > > > **Response to comment (continued)**
> > > > >
> > > > > # Summarizing the concerns
> > > > >
> > > > > * As claimed by the short horizon bias (SHB) paper, if LR hypergradients are computed by truncating the number of steps differentiated through (short rollout), this can lead to biased hypergradients. This form of HPO is therefore not necessarily a good strategy.
> > > > >
> > > > > * In particular, since our algorithm differentiates through a small number of FT steps in the online updates, your concern is that these hypergradients will be biased and not give an effective learning signal, and therefore this is a limitation of our algorithm.
> > > > >
> > > > >
> > > > > * Given our results improve on baselines, another question is why the types of hyperparameters we tune do not suffer from the issues that were seen in the SHB paper when optimizing the LR.
> > > > >
> > > > > # Response:
> > > > >
> > > > > ## On results already in our paper implying that bias is not necessarily an issue
> > > > >
> > > > > Your comment implies that the gradients we obtain through FT are biased, and therefore may not lead to effective learning.
> > > > >
> > > > > However, as showcased in our results (Table 1 and Table 2 in the main paper), setting K=1 can noticeably improve results over baselines. This is in contrast to one of the main points of the SHB paper, where optimizing over a short horizon could perform worse than a fixed hyperparameter (see Figure 7 in the SHB paper). We therefore believe that this problem of biased gradients due to short horizon rollouts is not necessarily a concern in our setting. Similar strategies to differentiate through one step of training were also employed in other meta-learning works (references [1-5] in our original response).
> > > > >
> > > > >
> > > > >
> > > > > ## New experiments studying the short horizon bias issue
> > > > >
> > > > > To further study whether we suffer from a problem of biased gradients in our method, we study two new experiments.
> > > > >
> > > > >
> > > > > ### Experiment 1: Toy setting comparing with BPTT
> > > > >
> > > > > Firstly, as you suggest, we consider a toy setting where PT and FT can fit into memory in order to study whether BPTT improves on our method. We return to our first synthetic setting of Section 4, where we aim to learn rotation augmentations for MNIST PT, given that the FT set is augmented in a specific way. In the following experiments, the FT set is augmented with rotations drawn from N(90,1).
> > > > >
> > > > > **Experimental setup:**
> > > > > * BPTT: The PT augmentation distribution is initialized to N(45,1).
> > > > > We do 500 steps of PT and 500 steps of FT steps, and use BPTT (through these 1000 optimization steps) to optimize the augmentations. This is near the limit of what we could fit into our GPU memory.
> > > > > * Meta-parameterized PT: We run our algorithm. The PT augmentation distribution is initialized to N(45,1). We set P=1, K=1, running for 500 PT and FT steps overall (for a fair comparison with BPTT).
> > > > > * Optimal augmentations: We set the PT augmentation distribution to be the optimal setting (i.e., identical to that used for FT): N(90,1).  This is also run for 500 PT and 500 FT steps.
> > > > >  * Initialization augmentations: We set the PT augmentation distribution to be: N(45,1) as a baseline. This is also run for 500 PT and 500 FT steps.
> > > > >
> > > > >
> > > > > **Results:**
> > > > >
> > > > > * BPTT without compute limitations: Running BPTT for 500 hyperparameter optimization steps takes about 20 hours. Doing so, it achieves a test accuracy of 88.0%.
> > > > > * Meta-parameterized PT: Running our method takes about 30 minutes. This achieves a test accuracy of 87.6%.
> > > > > * BPTT with compute limitations: Limiting the compute budget of BPTT to be similar to our method, it obtains a test accuracy 83.4%.
> > > > > * Optimal augmentations: This achieves a test accuracy of 88.3%.
> > > > > * Initialization augmentations: This achieves a test accuracy of 80.1%.
> > > > >
> > > > >
> > > > > **Analysis:**
> > > > >
> > > > > * As seen, our method, with about 2-3% of the compute time and significantly lower memory cost than BPTT, obtains very comparable performance in this toy domain, and almost matches the performance with the optimal hyperparameter setting.
> > > > > This indicates effective optimization of the hyperparameters.
> > > > > * This performance is achieved even when differentiating through a short FT optimization of 1 step.
> > > > >
> > > > >
> > > > > **Conclusions:**
> > > > >
> > > > > * In this toy domain, our method obtains performance very comparable to BPTT and the optimal hyperparameter setting, and has a fraction of the compute and memory cost of BPTT. This suggests that optimizing the augmentations online is not incurring significant short horizon/truncation bias.
> > > > > * In addition to these results, note that in Section 3 of our paper, we also show in two settings that our algorithm effectively recovers the optimal hyperparameters (augmentations and example weighting policies), indicating successful optimization, even with K=1.
> > > > > * Taken together, we believe our method is not suffering from significant SHB and can effectively optimize hyperparameters online.
> > > > >
> > > > > ### Experiment 2: K=1 vs an optimal FT head
> > > > >
> > > > > In this second study, instead of differentiating through the FT training (i.e., SGD/Adam optimization steps) to compute the hypergradient, we examine the following setup on the ECG SimCLR domain (Section 6 in the paper).
> > > > >
> > > > > * We frame the FT problem as a linear regression problem of predicting the true labels (this can be viewed as optimizing the model’s Brier score), using the pretrained representations as input.
> > > > >
> > > > > * This linear regression model has a closed form solution for the optimal parameters, and we compute these as the optimal FT parameters (similar to if we ran FT on a linear model for a large number of gradient steps).
> > > > > * This optimal setting is a function of the pretrained parameters and the hyperparameters, and we thus use this to compute the hypergradient.
> > > > >
> > > > > * We compare how this performs to differentiating through one step of training, as in Table 2 in our paper.
> > > > >
> > > > > **Results**
> > > > >
> > > > > As with the earlier results, the results are test AUC at different FT dataset sizes:
> > > > >
> > > > > |                          | 100       | 250       | 500       | 1000      |
> > > > > |--------------------------|-----------|-----------|-----------|-----------|
> > > > > | K=1(from Table 2)        | 76.1+-0.5 | 77.8+-0.4 | 81.7+-0.2 | 84.0+-0.3 |
> > > > > | Closed form linear model | 75.1+-0.7 | 77.5+-0.4 | 80.6+-0.3 | 84.4+-0.3 |
> > > > > | No meta-learning         | 74.6+-0.4 | 76.5+-0.3 | 79.8+-0.3 | 82.2+-0.3 |
> > > > >
> > > > >
> > > > > As is seen, our approach is competitive with or improves on this closed form linear model solution at all data regimes.
> > > > >
> > > > > While not an exact comparison, this does show that our FT gradients are as good or better than that obtained through an exact, closed-form solution to the FT problem, further indicating that we do not necessarily suffer from significant short-horizon bias.
> > > > >
> > > > >
> > > > >
> > > > > We hope we have demonstrated through the above experiments that our method does not suffer from the SHB problem in the experiments considered. However, we note that if our method did have worse performance due to truncating FT and optimizing online, that there exist methods to alleviate such issues (see reference [8] in our original comment).
> > > > >
> > > > >
> > > > > ## Why might short rollouts work better for us than for optimizing the LR?
> > > > >
> > > > >
> > > > > The learning rate directly affects the rate at which we approach the critical point, but it does not directly change the critical point.
> > > > > As seen in the SHB paper’s analysis, optimizing the LR with short rollouts results in (far too aggressively) decaying the step size to decrease variance and converge faster.
> > > > >
> > > > > In contrast, other hyperparameters, like weight decay or augmentations, directly change the fixed point that we are converging to (as opposed to just the rate).  In setups where the hyperparameters directly affect the fixed point, SHB has not been observed --- for example, see the follow-up papers in the references below [9-12], a subset of which directly feature the authors of the SHB paper. These papers do online, limited horizon optimization of hyperparameters directly affecting the critical point.
> > > > >
> > > > >
> > > > > ## On the relative impact of PT and FT components to the hypergradient
> > > > >
> > > > > We ran an additional experiment in our SimCLR domain (section 6) where we do *no* finetuning when optimizing the hyperparameters (K=0 FT steps): that is, we use a randomly initialized linear classifier on top of the pretrained representations when we compute the FT loss.
> > > > >
> > > > > The meta-learning problem here corresponds to learning PT hyperparameters that optimize the performance of a randomly initialized linear classifier.  The Jacobian $\frac{\partial \theta_{\text{FT}}^*}{\partial \theta_{\text{PT}}^*}$ will be the identity matrix. This experiment directly tests what happens when we have a noisy FT gradient, but an informative gradient through PT.
> > > > >
> > > > > ### Results
> > > > >
> > > > > As with the earlier results, the results are test AUC at different FT dataset sizes:
> > > > >
> > > > >
> > > > > |                     | 100         | 250         | 500         | 1000        |
> > > > > |---------------------|-------------|-------------|-------------|-------------|
> > > > > | K=1(from table 2)   | 76.1+-0.5   | 77.8+-0.4   | 81.7+-0.2   | 84.0+-0.3   |
> > > > > | Random FT component | 75.3+-0.5   | 77.1+-0.5   | 80.5+-0.4   | 83.7+-0.3   |
> > > > > | No meta-learning    | 74.6+-0.4 | 76.5+-0.3 | 79.8+-0.3 | 82.2+-0.3 |
> > > > >
> > > > >
> > > > > As can be seen, unrolling through one step of FT improves upon using a noisy FT gradient in all cases. Using a random FT component but an informative PT component (second row) improves on not optimizing the augmentations at all. This implies that both the PT and FT components inform the optimization of the augmentations.
> > > > >
> > > > > ## References
> > > > >
> > > > > [9] Learning to reweight examples for robust deep learning, ICML 2018
> > > > >
> > > > > [10] Self-Tuning Networks: Bilevel Optimization of Hyperparameters using Structured Best-Response Functions, ICLR 2018
> > > > >
> > > > > [11] Graph HyperNetworks for Neural Architecture Search, ICLR 2018
> > > > >
> > > > > [12] Delta-STN: Efficient Bilevel Optimization for Neural Networks using Structured Response Jacobians, arXiv 2020

---

> > > > > ### Comment · Reviewer_BiAi · 2021-08-23
> > > > > **Reply 2**
> > > > >
> > > > > Thank you for your additional experiments and arguments. I am slightly less skeptical of your online FT component after your explanations. I have increased my score to reflect this. I have also lowered my confidence to reflect the fact that I am not confident I understand how good/significant the baselines in your experiment section are.

---

> > > > > > ### Author Response · Authors · 2021-08-24
> > > > > > **Thank you!**
> > > > > >
> > > > > > Thank you for your time, and for looking over our new experiments and for revising your score.

---

> ### Author Response · Authors · 2021-08-23
> **Author note to reviewer**
>
> Dear Reviewer,
>
> Thank you once again for your review and for your follow-up comment. In our first response and our follow-up comments, we believe we have replied to your main points regarding: the short horizon bias, significance of the datasets and baselines considered, novelty of our method, toy experiment setup, and clarity. We would greatly appreciate it if you could update your review, given our response.
>
> Thank you again for your time!

---

### Official Review · Reviewer_UJ1Z · 2021-07-17

**Rating:** 5
**Confidence:** 5

**Summary:**

This paper propose to solve a very important problem, meta-learning hyperparameters in pre-training stage, followed by fine-tuning a target task. Considering that the hyperparameters can be high-dimensional, the authors consider gradient-based hyperparameter optimization (HO) methods such as IFT based methods and unrolled differentiation. The main difficulty comes from the huge computational cost of dealing with long PT and FT trajectory and repeating such meta-optimization steps. Authors thus simply reduce the number of gradient steps used for each PT and FT stage. Yet, the experimental results demonstrate the effectiveness of the proposed methods over some of the simple baselines.

**Limitations And Societal Impact:**

The authors adequately adressed their limitations and social impract.

**Main Review:**

== Pros ==

1. I think the paper tackles the very important and interesting problem, meta-learning of PT hyperparameters followed by FT. As far as I know, there are few literatures that explicitly tackle this problem. The problem is challenging because both PT and FT involve long optimization trajectory in practice, leading the optimization problem computationally very expensive. Therefore, I would say that the motivation of this paper is very clear and important, in terms of extending the current range of meta-learning and hyperparameter-optimization study.

2. The paper is well written and easy to understand, combining the two most popular techniques for HO (IFT and unrolled diff.).

3.  The experimental results show that although the proposed method is simple and straightfoward, it could gain some improvements over the baselines they considered.


== Cons ==

1. As far as I understand, the main difficulty should come from the long PT and FT optimization trajectory used in practice (e.g. each at least thousands of SGD steps). However, the authors simply set each of them to very short number of steps, such as $P=10$ and $K=1$ for both of the real-world experiments. It will lead to short horizon bias, with the solution being less appealing. For example, Shin et al. recently proposed a method that can increase the frequency of meta-updates even with long inner-optimization trajectories. It would be better if the authors could think of some ways to deal with the short horizon bias in any way, because exactly that is the main challenge of this problem.

2. The proposed method is fairly straightforward to think of, combining the two existing techniques for HO and unrolled optimization (MAML). I agree that the method is reasonable, but not sure how much it is technically contributing. Also, as mentioned in section 8 Scopes and Limitations, IFT based methods cannot handle hyper-parameters that do not affect the PT optimization landscape.

3. In the experiments, I think the baselines are too few. Specifically, there are no baselines that can learn the hyperparameter $\phi$. What if we learn $\phi$ only with the PT dataset based on conventional HO framework? We may split the $D_\text{PT}$ into $D_\text{PT}^\text{train}$ and $D_\text{PT}^\text{val}$, and optimize $\phi$ with Neumann IFT method as you did. I expect it will work well because the authors reported good performance on Partial FT-data setting, where we only use a fraction of FT tasks for meta-training and meta-test with the exclusive set of FT tasks no seen during the meta-training. It means that whatever tasks we use for learning $\phi$, the learned $\phi$ will generalize well to unseen tasks. If it works well, then the importance of meta-learning over PT-FT framework will become questionable.

4. (minor comment) There is no qualitative analysis. It would be interesting to visualize the learned $\phi$ and give the readers some intuition how it helped with the performance (or any other visualization).

5. (minor comment) Missing reference for meta-learning of self-supervised learning : Kang et al.

= References =
- Shin et al., Large-Scale Meta-Learning with Continual Trajectory Shifting, ICML 2021
- Kang et al., Neural Mask Generator: Learning to Generate Adaptive Word Maskings for Language Model Adaptation, EMNLP 2020



**Time Spent Reviewing:**

6 hours

---

> ### Author Response · Authors · 2021-08-10
> **Author's response to review (part 1)**
>
> We thank you for your comments and time!
>
> Below, we provide detailed responses to each of the 5 cons identified in your review. In summary, we
>
> 1. Clarify that our method works in an online manner, so actually runs a significant number of PT/FT steps in total, and further provide novel results using larger values of $K$ to assess the impact this parameter has on our results.
>
> 2. Clarify the technical significance of our method.
>
> 3. Provide details of two meta-learning baselines which we compare against (one of which is currently in the appendix of the paper and the other is new).
>
> 4. Pointed out the existence of qualitative analyses in the appendix.
>
> 5. Note that we will include the suggested references.
>
>
> We look forward to hearing your response to these key points. If you feel we have not sufficiently addressed your concerns to motivate increasing your score, we would love to hear from you further on what points of concern or confusion remain and how we can improve the work in your eyes. Thank you again!
>
>
> # Detailed Response
>
>
> ## Con 1: On the number of PT and FT steps & the short horizon bias
>
> #### **Re:** *“... the main difficulty should come from the long PT and FT optimization trajectory used in practice (e.g. each at least thousands of SGD steps). However, the authors simply set each of them to very short number of steps, such as P=10 and K=1 for both of the real-world experiments.”*
>
>
> We apologize for any confusion on this point. Our method actually does not reduce the number of steps for PT & FT in general. As stated in Algorithm 1, we solve the hyperparameter optimization problem in an online fashion, similarly to related work [1,2,3]. In this way, while we do use a small number of PT & FT steps in each inner loop iteration, the total number of steps used during PT and FT is therefore large for both domains, on the order of $10^4$ to $10^5$ steps (lines 212-214, lines 287-292). This online approach is significantly more computationally efficient than performing offline updates.
>
>
> #### **Re:** *“[the short number of steps] will lead to short horizon bias... It would be better if the authors could think of some ways to deal with the short horizon bias in any way.”*
>
>
> Thank you for raising this important concern -- it warranted more commentary in the main text and we will work to improve our clarity on this point.
>
> **In summary:** We understand short-horizon bias to be a special case of the bias induced by truncating telescoping sums for optimization parameters.  The effects of the truncation can be pronounced with optimization parameters, but there exist methods like [7] to deal with these if they occur.  Prior works of [1-5] did not observe significant truncation bias in similar settings to ours, so we also did not expect this bias to be a significant issue.  We have added additional experiments below to investigate if we have a truncation bias, which show the effect is limited.
>
>
> **A closer look:** There are two hypergradients in our system that could suffer from bias: the PT hypergradients and the FT hypergradients. *In both cases, we feel that any impact from biased hypergradients is minimal, and that the significant performance improvements observed still demonstrate the value of our algorithm.*
>
> We will argue for this claim through each hypergradient term separately.
>
>
> **PT Hypergradient:** The PT hypergradient does not suffer from the short-horizon bias because the PT model is expected to have approximately converged at each hyperparameter update. This is not only a requirement of the implicit function theorem and the algorithm from [1] to apply, but also is directly enforced in our system through the use of online-updates and a warmup period (see full algorithm description in Appendix B of the paper: Algorithm 2, and lines 598-601). As seen in prior work using this method [1-3], computing hypergradients in such an online fashion results in effective learning, so we do not believe this to be a significant issue.
>
>
>
> **FT Hypergradient:** For the gradient through FT, we acknowledge that differentiating through only one step could, in theory, produce biased hypergradients.
>
>
> However, prior work on meta-learning structures similar to ours has computed meta-parameter updates in a similar fashion by differentiating through one step of training, and has obtained impressive, near SOTA results [4], [5]. Like these works, we also obtain impressive improvements even with $K=1$.
>
> If short-horizon bias is an issue, then we could use the method from [7] to fix it.  We did not expect there to be a significant bias here, due to [4], [5], but in the interest of ensuring this bias is not a significant concern, we have added experiments showing that the horizon does not significantly affect the result.
>
> In particular, we considered the meta-parameterized SimCLR and ECG data experiments (Section 6) and studied using a larger number of FT optimization steps $K$, namely $K=5$ and $K=10$. The results are as follows, including the results in the paper for $K=0$ (no meta-learning) and $K=1$ for completeness, from Table 2.  Each column of results is the test AUC at different FT dataset sizes.
>
>
>
> |                        	|    100    	|     250    	|    500    	|    1000   	|
> |------------------------	|:---------:	|:----------:	|:---------:	|:---------:	|
> | K=0 (No meta-learning) 	| 74.6+-0.4 	| 76.5+-0.3  	| 79.8+-0.3 	| 82.2+-0.3 	|
> | K=1                    	| 76.1+-0.5 	| 77.8+-0.4  	| 81.7+-0.2 	| 84.0+-0.3 	|
> | K=5                    	| 76.6+-0.2 	| 78.3+-0.3  	| 81.9+-0.4 	| 84.2+-0.2 	|
> | K=10                   	| 76.3+-0.5 	| 78.1 +-0.4 	| 81.7+-0.4 	| 84.3+-0.2 	|
>
>
>
>
>
> As seen from the results, we do observe some improvement with more steps, but diminishing returns as $K$ increases further.
>
> To summarize, we see that:
> 1. Most of the improvement over the no meta-learning baseline appears to come from just setting $K=1$;
>
> 2.  Running with more $K$ is certainly possible with our algorithm given we only need to unroll the network head;
>
> 3.  Prior work has shown the value of single-step differentiation to compute meta-parameter gradients;
>
> Given all this, we do not see this as a major limitation of our method.
>
>
>
> ## Con 2: On the Novelty of our Method
>
> #### **Re:** *”The proposed method is fairly straightforward to think of.... not sure how much it is technically contributing.”*
>
>
> We respectfully disagree with the point that our method is too straightforward to be a technical contribution. We believe that the simplicity of the method is a strength -- it is an intuitive way of combining two existing ideas, is well-justified and theoretically motivated, and achieves strong performance on two important real-world domains. Furthermore, to the best of our knowledge, despite its intuitive nature, there is no prior work studying this particular approach, so we do view this as an important contribution.
>
>
> #### **Re:** *“Also, as mentioned in section 8 Scopes and Limitations, IFT based methods cannot handle hyper-parameters that do not affect the PT optimization landscape.”*
>
> We agree that handling other forms of hyperparameters such as learning rates are an opportunity for future work.
>
> However, we feel that the ability to handle important hyperparameters such as task weighting schemes, data augmentation networks, and more (as evidenced by our real-world experiments) is still of significant value. Moreover, our approach is quite flexible in that it could theoretically apply to any differentiable hyperparameters/models. There are a large range of other hyperparameters that could be optimized with our approach, including: auxiliary labels [3], learned attention masks [2], and high-dimensional regularization parameters [1]. Given all this, we do not feel this limitation is sufficient to motivate rejection.

---

> > ### Author Response · Authors · 2021-08-10
> > **Author's response to review (part 2)**
> >
> > ## Con 3: On Baselines
> >
> > #### **Re:** *“I think the baselines are too few. Specifically, there are no baselines that can learn the hyperparameter $\phi$”*
> >
> >
> > To address this concern, we would like to highlight a hyperparameter tuning baseline that we already consider in the appendix, a competitive multitask learning baseline we already consider, and introduce a new hyperparameter tuning baseline to address the reviewer’s concerns specifically. *In all three cases, our method significantly outperforms these baselines, thereby demonstrating its value over competing techniques.*
> >
> > **Existing Baselines:** In the appendix,  we consider a baseline ‘CoTrain + Learned Weights’ for GNN PT that applies the method of Lorraine et. al [1] to learn weights over the many different tasks. Please see Appendix E.2 for further details, and quantitative results for this method in Tables 5-7, Appendix E. We find that our approach significantly outperforms this hyperparameter learning baseline, and also study a potential reason for this in Table 6. While this is not the exact baseline the reviewer proposes, it does directly test whether meta-learning over PT and FT together is relevant.
> >
> > We also compare to a competitive multitask learning baseline, namely CoTrain + PCGrad [6], leveraging a recently published method for improved multitask learning. Details are in Lines 209-211 and Appendix E.2 and results in Tables 1,4,5,7,8. Again, we find that our proposed method improves on this baseline.
> >
> > **New Baseline:** As another example of a meta-learning baseline, for the SimCLR ECG domain, we studied a situation where we:
> >
> > * Learn the augmentation parameters for supervised learning on the finetuning set $\mathcal{D}_{\text{FT}}$, following the method from [1].
> >
> > * Perform SimCLR PT with these learned augmentation parameters
> >
> > * Perform fine-tuning, and then evaluate on the FT test set.
> >
> > Note that this approach could be considered an ablation where we learn augmentations for a traditional supervised learning problem and see how these compare to augmentations directly optimized for semi-supervised learning, over a two-stage learning problem.
> >
> > The results are as follows. Again, each column of results is the test AUC at different FT dataset sizes.
> >
> >
> >
> > |                                  	|    100    	|    250    	|     500    	|    1000   	|
> > |----------------------------------	|:---------:	|:---------:	|:----------:	|:---------:	|
> > | Meta-Parameterized SimCLR        	| 76.1+-0.5 	| 77.8+-0.4 	| 81.7+- 0.2 	| 84.0+-0.3 	|
> > | Hyperparameter learning baseline 	| 74.6+-0.6 	| 77.0+-0.3 	| 79.6+-0.4  	| 82.8+-0.2 	|
> >
> >
> >
> > As can be seen, learning hyperparameters over the PT and FT stages together improves performance.
> >
> >
> >
> > ## Con 4: On qualitative analysis
> >
> > #### **Re:** *“There is no qualitative analysis. It would be interesting to visualize the learned ϕ and give the readers some intuition how it helped with the performance (or any other visualization).”*
> >
> > We provide qualitative analysis on the representations learned by the graph networks with different PT strategies using representational analysis tools (Appendix E, Figure 3), consider the histogram of learned weights for two approaches (Appendix E, Figure 4), and an analysis on negative transfer (Appendix E, Figure 5).
> >
> >
> > ## Con 5: Missing reference
> > Thank you for providing this. We will add this to our related work.
> >
> >
> > ## References:
> >
> > [1] Optimizing Millions of Hyperparameters by Implicit Differentiation, AISTATS 2020
> >
> > [2] Teaching with Commentaries, ICLR 2021
> >
> > [3] Auxiliary Learning by Implicit Differentiation, ICLR 2021
> >
> > [4] Meta Pseudo Labels, CVPR 2021
> >
> > [5] Learning Data Manipulation for Augmentation and Weighting, NeurIPS 2020
> >
> > [6] Gradient Surgery for Multi-Task Learning, NeurIPS 2020
> >
> > [7] Efficient Optimization of Loops and Limits with Randomized Telescoping Sums, ICML 2019

---

> ### Author Response · Authors · 2021-08-23
> **Author note to reviewer**
>
> Dear Reviewer,
>
> Thank you once again for your review. In our response, we believe we have replied to your main points regarding: the number of PT/FT steps and the short horizon bias, novelty of our method, baselines, qualitative analysis, and the missing reference. We would greatly appreciate it if you could update your review, given our response.
>
> Thank you again for your time!

---

### Author Response · Authors · 2021-08-10
**Author's overall response to reviews**

We thank all the reviewers for their time and comments on our paper. We are encouraged to see that the reviewers found the problem setting we considered to be interesting, and generally found the paper clear.

We respond to each reviewer’s comments individually, and outline some of the key points discussed here:


1. **Simplicity, novelty, and technical contribution:** We believe that the simplicity of our method is a strength -- it is an intuitive way of combining two existing ideas, is well-justified and theoretically motivated, and achieves strong performance on two important real-world domains. Furthermore, to the best of our knowledge, despite its intuitive nature, there is no prior work studying this particular approach, so we do view this as an important contribution.



2. **Optimization trajectory lengths and short-horizon bias:** We give new results studying this phenomenon further, clarify some points in our algorithm regarding this concern, and point to previously published meta-learning work that performs effectively even when differentiating through shorter trajectories.


3. **Datasets and Baselines:** We provide more clarity on the significance of the datasets and the various baselines (e.g., standard and improved multi-task learning and meta-learning baselines) that we consider in the paper and point to the specific results. We also provide results for a new meta-learning baseline we considered in the semi-supervised learning setting, which we improve on with our method.


4. **Computational questions:** We provide more information on time/memory requirements for our method.

---

### Decision · Program_Chairs · 2021-09-27

**Decision:**

Accept (Poster)

**Comment:**

Pretrain-finetuning is an important and popular learning framework for modern deep learning models, and has achieved significant success. How to learn some hyperparameter of the framework is important yet not well explored. This paper proposes a meta learning strategy for this via direct and implicit differentiation to perform gradient descent. The technique is not new but has not been applied to this problem.

All reviewers seem to consider the paper to be in the borderline. They raised a number of questions including the setting of the number of PT/FT steps, if it will cause short horizon bias, and also asked for some additional baselines to demonstrate the effectiveness of the proposed method. Most of the questions, in my opinion, have been well addressed in the rebuttal with further explanation and extra experiments. Thus I would recommend acceptance of the paper. One concern I have is that it seems the scale of the data in the experiments is not large enough (from what I understand in the paper, though the statistics of some data have not been provided). I guess investigating the large-scale setting is important because this is where the PT/FT framework really reveals its power. I would recommend the authors to revise the paper according to the comments from the reviewers, and consider extra large-scale experiments (such as the ImageNet scale data) to make the paper stronger.